# To Federate or Not To Federate: Incentivizing Client Participation in Federated Learning

## Abstract

Federated learning (FL) facilitates collaboration between a group of clients who seek to train a common machine learning model without directly sharing their local data. Although there is an abundance of research on improving the speed, efficiency, and accuracy of federated training, most works implicitly assume that all clients are willing to participate in the FL framework. Due to data heterogeneity, however, the global model may not work well for some clients, and they may instead choose to use their own local model. Such disincentivization of clients can be problematic from the server's perspective because having more participating clients yields a better global model, and offers better privacy guarantees to the participating clients. In this paper, we propose an algorithm called INCFL that explicitly maximizes the fraction of clients who are incentivized to use the global model by dynamically adjusting the aggregation weights assigned to their updates. Our experiments show that INCFL increases the number of incentivized clients by 30-55% compared to standard federated training algorithms, and can also improve the generalization performance of the global model on unseen clients.

## 1 Introduction

Federated learning (FL) is a distributed learning framework that enables the training of a machine learning model using a network of clients (e.g., mobile phones, hospitals) [1], without having to transfer the clients' data to a central server. In the standard FL framework [1–6], clients perform a few updates using their local data, and a central server aggregates these updates into a single global model. As the global model is based on the union of all the local datasets, it is expected to generalize well for the entire client population. However, due to heterogeneity of the data between clients [7], not all clients stand to benefit from federation. While FL can produce a model that performs well on average, for some clients, it may perform even worse than a model trained in isolation on their limited local data. Our experiments (Section 4) demonstrate that local models trained in isolation on FL benchmarks can indeed outperform global models obtained by commonly used FL algorithms.

When a client participates in FL, it incurs the cost of contributing its local data and computational resources to the federation in return for receiving a global model. However, if a local model trained in isolation is better than the global model, the client may not be incentivized to participate in FL—causing it to opt out of contributing to and using the global model in the future. This lack of incentives for clients to participate in FL can be problematic from the server's perspective. Having a large pool of clients willing to participate in training is beneficial, if not imperative, to ensure the performance of FL models [8, 9]. When a large number of clients participate, the global model is based on a larger pool of data, allowing better generalization to new clients that may join in the future. Having a larger number of participating clients can also improve privacy-utility trade-offs by mitigating the impact of each individual client on the global model [10–12].

Submitted to 36th Conference on Neural Information Processing Systems (NeurIPS 2022). Do not distribute.

In this work, we seek to answer the following pertinent question: *How can we actively incentivize clients to use and contribute to a federated global model, rather than training local models in isolation?* To address this question, we propose an algorithm called INCFL to train a global model that dramatically improves the fraction of incentivized clients in comparison to standard FL algorithms. Our key contributions are summarized as follows.

- In Section 2 we formalize the notion of client incentives by defining a metric called the incentivized participation rate (IPR), which measures the fraction of clients willing to participate in the FL framework. We propose to maximize a sigmoid relaxation of the IPR, which makes the objective differentiable and enables the use of common gradient-based optimization algorithms.

- In Section 3 we propose a federated algorithm called INCFL to maximize incentivized client participation. INCFL dynamically adjusts the weight assigned to each client's local update when aggregating the updates at the central server. The method allows for partial client availability for training, it is applicable to general non-convex objectives (with convergence guarantees), and it is stateless (does not require clients to maintain local parameters during training).

- In Section 4, we empirically validate the performance of INCFL by comparing it with standard FL algorithms for multiple data sets. INCFL is able to increase the number of incentivized clients by 30-55%, and also ensures that the global model generalizes well to unseen clients.

As surveyed in [13], previous works investigating client incentives in FL have typically done so from a game-theoretic perspective and for toy problems such as mean estimation. In contrast, our work is generally applicable to non-convex objectives, and considers a server that seeks to train a single global model that will be preferred by the maximum number of clients, thus incentivizing them to participate in FL. We provide a more detailed review of prior work in Appendix A below.

## 2 Problem Formulation

We consider a FL setup where $M$ clients are connected to a central server. For each client $k \in \{1, 2, \ldots, M\}$, its true local loss function is given by $f_k(\mathbf{w}) = \mathbb{E}_{\xi \sim \mathcal{D}_k}[\ell(\mathbf{w}, \xi)]$ where $\mathcal{D}_k$ is the true data distribution of client $k$, and $\ell(\mathbf{w}, \xi)$ is the composite loss function for the model $\mathbf{w} \in \mathbb{R}^d$ for data sample $\xi$. In practice, each client only has access to its local training dataset $\mathcal{B}_k$ with $|\mathcal{B}_k| = N_k$ data samples sampled from $\mathcal{D}_k$. Client $k$'s empirical local loss function is $F_k(\mathbf{w}) = \frac{1}{|\mathcal{B}_k|} \sum_{\xi \in \mathcal{B}_k} \ell(\mathbf{w}, \xi)$. Our setup is applicable to both cross-device and cross-silo FL as we do not make any specific assumptions about the nature of the clients or their constraints.

**Defining Client Incentives in FL.** The goal of each client is to find a model that minimizes its true loss function, which we denote as $\mathbf{w}_k^* := \mathrm{argmin}_{\mathbf{w}} f_k(\mathbf{w})$. To do so, we consider that each client does some solo training on its local dataset $\mathcal{B}_k$ to obtain an approximate local model $\widehat{\mathbf{w}}_k$. For example, $\widehat{\mathbf{w}}_k$ can be found by running a few steps of SGD on the empirical loss $F_k(\mathbf{w})$. Since the local dataset size is in general small, $\widehat{\mathbf{w}}_k$ may not generalize well to the true distribution $\mathcal{D}_k$ of a client. Therefore we say that a client is incentivized to participate in FL (i.e., use the federated model) if the federated model gives better generalization performance than its local model.

**Definition 1** (Client Incentive). *Given a global model* $\mathbf{w}$, *client* $k \in [M]$ *is said to be incentivized to participate in FL if* $f_k(\mathbf{w}) < f_k(\widehat{\mathbf{w}}_k)$, *that is, the global model is better than its own local model.*

In practice, clients can have a separate validation dataset on which they compare the losses of the global model and their local model to decide if they are incentivized to participate. In general, $f_k(\widehat{\mathbf{w}}_k)$ in Definition 1 acts as a performance benchmark for the global model and can also be replaced by a different value depending on the specific need of a client.

**Standard FL Objective does not account for Client Incentives.** In standard FL, clients collaboratively minimize the objective $F(\mathbf{w}) = \sum_{k=1}^{M} p_k F_k(\mathbf{w})$, where the aggregation weights $p_k$ are usually set as $p_k \propto |\mathcal{B}_k|$. Observe that this objective function does not consider client incentives as defined in Definition 1 and implicitly assumes that all clients will participate in training and use the global model. However, due to clients' data heterogeneity, this assumption may not hold in general.

**Incentivized Participation Rate (IPR).** Based on Definition 1, we formulate the following metric to explicitly measure the fraction of clients that are incentivized for a given federated global model $\mathbf{w}$:

$$\text{Incentivized Participation Rate (IPR)} = \frac{1}{M} \sum_{k=1}^{M} \mathbb{I}\{f_k(\mathbf{w}) < f_k(\widehat{\mathbf{w}}_k)\}, \tag{1}$$

where $\mathbb{I}$ is the indicator function. Note that IPR only looks at whether or not a client is incentivized and not *how much* a client is incentivized (or disincentivized) since the decision to participate in FL is binary. Another variation of (1) could be to measure the incentive margin of clients, e.g. $\sum_k \max\{f_k(\widehat{\mathbf{w}}_k) - f_k(\mathbf{w}), 0\}$, but this does not capture the motivation behind our work which is to improve the number of the incentivized clients in FL. To the best of our knowledge, a similar indicator based metric has not been explored previously in the FL literature.

## 2.1 Proposed INCFL Objective

A naïve approach to increase the number of incentivized clients with our definition of client incentives in (1) is directly maximizing the IPR as follows:

$$\max_{\mathbf{w}} \left[ \frac{1}{M} \sum_{k=1}^{M} \mathbb{I}\{f_k(\mathbf{w}) < f_k(\widehat{\mathbf{w}}_k)\} \right] = \min_{\mathbf{w}} \left[ \frac{1}{M} \sum_{k=1}^{M} \text{sign}(f_k(\mathbf{w}) - f_k(\widehat{\mathbf{w}}_k)) \right]. \tag{2}$$

where $\text{sign}(x) = 1$ if $x \geq 0$ and 0 otherwise. There are two immediate difficulties in minimizing (2). First, clients may not know their true data distribution $\mathcal{D}_k$ to compute $f_k(\mathbf{w}) - f_k(\widehat{\mathbf{w}}_k)$. Secondly, the sign function makes the objective nondifferentiable and limits the use of common gradient-based methods. We resolve these issues by proposing a "proxy" for (2) with the following relaxations.

1. **Replacing the Sign function with the Sigmoid function** $\sigma(\cdot)$ **[14]:** Replacing the non-differentiable 0-1 loss with a smooth differentiable loss is a standard tool used in optimization [15, 16]. Given the many candidates (e.g. hinge loss, ReLU, sigmoid), we find that using the sigmoid function is essential for our objective to faithfully approximate the true objective in (2). We discuss theoretical implications of using the sigmoid loss in more detail in Appendix B.1.

2. **Replacing** $\sigma(f_k(\mathbf{w}) - f_k(\widehat{\mathbf{w}}_k))$ **with** $\sigma(F_k(\mathbf{w}) - F_k(\widehat{\mathbf{w}}_k))$**:** As clients do not have access to their true distribution $\mathcal{D}_k$ to compute $f_k(\cdot)$ we propose to use an empirical estimate $\sigma(F_k(\mathbf{w}) - F_k(\widehat{\mathbf{w}}_k))$. Since $\widehat{\mathbf{w}}_k$ is locally trained, it is likely that $F_k(\widehat{\mathbf{w}}_k) < f_k(\widehat{\mathbf{w}}_k)$. On the other hand, the global model $\mathbf{w}$ is trained on the data of all clients, making it unlikely to overfit to the local data of any particular client, leading to $f_k(\mathbf{w}) \approx F_k(\mathbf{w})$ (see Appendix F.2). Hence, in most cases we have $f_k(\mathbf{w}) - f_k(\widehat{\mathbf{w}}_k) < F_k(\mathbf{w}) - F_k(\widehat{\mathbf{w}}_k)$ and since sigmoid is an increasing function, this implies that $\sigma(f_k(\mathbf{w}) - f_k(\widehat{\mathbf{w}}_k)) < \sigma(F_k(\mathbf{w}) - F_k(\widehat{\mathbf{w}}_k))$. Therefore, with this relaxation we are effectively trying to minimize an upper bound on our true objective.

With these relaxations, we present our proposed INCFL objective:

$$\text{INCFL Obj.} : \quad \min_{\mathbf{w}} \widetilde{F}(\mathbf{w}) = \min_{\mathbf{w}} \frac{1}{M} \sum_{i=1}^{M} \widetilde{F}_i(\mathbf{w}), \text{where } \widetilde{F}_i(\mathbf{w}) := \sigma(F_i(\mathbf{w}) - F_i(\widehat{\mathbf{w}}_i)). \tag{3}$$

Our experimental results in Section 4 support our intuition of these relaxations and convincingly demonstrate that minimizing our proposed objective leads to a much higher IPR than the standard FL objective. Before discussing the details of how we minimize our objective, we take a closer look at how our objective behaves for a mean estimation problem.

## 3 Proposed INCFL Algorithm

With the sigmoid approximation of the sign loss and for differentiable $F_k(\mathbf{w})$, our objective $\widetilde{F}(\mathbf{w})$ in (3) is differentiable and can be minimized with gradient descent and variants. Its gradient is given by:

$$\nabla \widetilde{F}(\mathbf{w}) = \frac{1}{M} \sum_{k=1}^{M} \underbrace{(1 - \widetilde{F}_k(\mathbf{w})) \widetilde{F}_k(\mathbf{w})}_{\text{aggregating weight}:=q_k(\mathbf{w})} \nabla F_k(\mathbf{w}). \tag{4}$$

Observe that $\nabla \widetilde{F}(\mathbf{w})$ is a **weighted aggregate** of the gradients of the clients' empirical losses, similar in spirit to the gradient $\nabla F(\mathbf{w})$ in standard FL. The key difference is that in INCFL, the weights $q_k(\mathbf{w}) := (1 - \widetilde{F}_k(\mathbf{w}))\widetilde{F}_k(\mathbf{w})$ are *incentive-dependent*, and are dynamically updated based on the current model $\mathbf{w}$, as we discuss below.

**Behavior of the Aggregation Weights** $q_k(\mathbf{w})$. For a given $\mathbf{w}$, the behavior of the aggregation weights $q_k(\mathbf{w})$ depend on the *empirical incentive gap*, $F_k(\mathbf{w}) - F_k(\widehat{\mathbf{w}}_k)$ (see Fig. 1) since $\widetilde{F}_k(\mathbf{w}) = \sigma(F_k(\mathbf{w}) - F_k(\widehat{\mathbf{w}}_k))$. When $F_k(\mathbf{w}) \ll F_k(\widehat{\mathbf{w}}_k)$, it implies that the global model $\mathbf{w}$ performs much better than the local model $\widehat{\mathbf{w}}_k$ at client $k$. Hence INCFL sets $q_k(\mathbf{w}) \approx 0$ to focus on the updates of other clients. Similarly if $F_k(\mathbf{w}) \gg F_k(\widehat{\mathbf{w}}_k)$, INCFL will set $q_k(\mathbf{w}) \approx 0$. This is because $F_k(\mathbf{w}) \gg F_k(\widehat{\mathbf{w}}_k)$ implies that the current model $\mathbf{w}$ is incompatible with the local model $\mathbf{w}_k$ at client $k$ and hence it is better to avoid optimizing for this client at the risk of disincentivizing other clients. INCFL gives the highest weight to those clients for which the global model performs similar to their local models, i.e. $F_k(\mathbf{w}) \approx F_k(\widehat{\mathbf{w}})$, since this allows it to increase IPR without hurting other clients' performance.

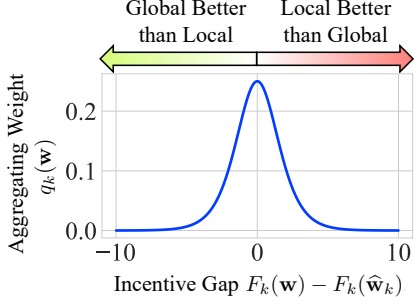

Figure 1: Aggregating weight $q_k(\mathbf{w})$ for any client $k$ versus the emprical incentive gap $F_k(\mathbf{w}) - F_k(\widehat{\mathbf{w}}_k)$. The weight $q_k(\mathbf{w})$ is small for clients that already have a very large incentive (global much better than local) or no incentive at all (local much better than global), and is highest for clients that are moderately incentivized (global similar to local).

**A Practical INCFL Solver.** Directly minimizing the INCFL objective using gradient descent can be slow to converge and impractical since it requires all clients to be available for training. Instead, we propose a practical INCFL algorithm, which uses multiple local updates at each client to speed up convergence as done in standard FL [1] and allow for partial client availability. We replace the gradient $\nabla F_k(\mathbf{w})$ with the *local update* $\Delta \mathbf{w}_k$ at a client, and aggregate these updates only from clients that are available in that round.

Let us use the superscript $(t, r)$ to denote the communication round $t$ and local iteration index $r$. At each round $t$, the server selects a new set of clients $\mathcal{S}^{(t,0)}$ uniformly at random and sends the most recent global model $\mathbf{w}^{(t,0)}$ to clients in $\mathcal{S}^{(t,0)}$. The clients in $\mathcal{S}^{(t,0)}$ then perform $\tau$ local iterations with local learning rate $\eta_l$ to calculate their updates as follows:

$$\text{Perform Local SGD: } \mathbf{w}_k^{(t,r+1)} = \mathbf{w}_k^{(t,r)} - \eta_l \mathbf{g}(\mathbf{w}_k^{(t,r)}, \xi_k^{(t,r)}) \quad \text{for all } r \in \{0, \dots, \tau - 1\}, \quad (5)$$

$$\text{Compute Local Update: } \Delta \mathbf{w}_k^{(t,0)} = \mathbf{w}_k^{(t,\tau)} - \mathbf{w}_k^{(t,0)}, \quad (6)$$

where $\mathbf{g}(\mathbf{w}_k^{(t,r)}, \xi_k^{(t,r)}) = \frac{1}{b} \sum_{\xi \in \xi_k^{(t,r)}} \nabla f(\mathbf{w}_k^{(t,r)}, \xi)$ is the stochastic gradient computed using a mini-batch $\xi_k^{(t,r)}$ of size $b$ that is randomly sampled from client $k$'s local dataset $\mathcal{B}_k$. The weight $q_k(\mathbf{w}_k^{(t,0)})$ can be computed at each client by calculating the loss over its training data with $\mathbf{w}_k^{(t,0)}$ which is a simple inference step. Clients in $\mathcal{S}^{(t,0)}$ then send back their local updates $\Delta \mathbf{w}_k^{(t,0)}$ and weights $q_k(\mathbf{w}_k^{(t,0)})$ to the server which updates the global model as follows:

$$\text{Global Update Rule: } \mathbf{w}^{(t+1,0)} = \mathbf{w}^{(t,0)} - \eta_g^{(t,0)} \sum_{k \in \mathcal{S}^{(t,0)}} q_k(\mathbf{w}^{(t,0)})\Delta \mathbf{w}_k^{(t,0)}, \quad (7)$$

where $\eta_g^{(t,0)} = \frac{\eta_g}{\sum_{k \in \mathcal{S}^{(t,0)}} q_k(\mathbf{w}^{(t,0)})+\epsilon}$ is the adaptive server learning rate with a fixed global learning rate $\eta_g$ and constant $\epsilon > 0$. We discuss the reasoning for such an adaptive learning rate along with the pseudo code and convergence bounds of our INCFL in Appendix C.

# 4 Experimental Results

We evaluate INCFL in four different settings: (i) logistic regression on a synthetic federated dataset (`Synthetic(1,1)` [2]), (ii) MLP trained on non-iid partitioned FMNIST [17], (iii) CNN trained on non-iid partitioned CIFAR10 [18], and (iv) MLP for sentiment classification trained on Sent140 [19]. We compare INCFL with well-known stateless FL algorithms that train a single model such as

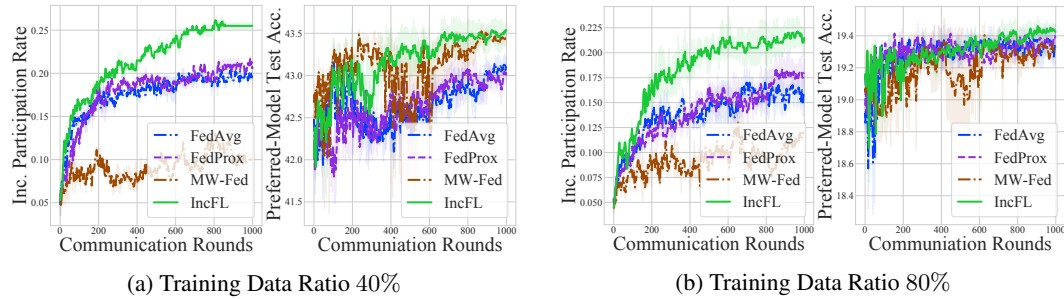

(a) Training Data Ratio 40%     (b) Training Data Ratio 80%

Figure 2: Incentivized participation rate (IPR), i.e., the fraction of incentivized clients, and preferred-model test accuracy evaluated on the test data for the synthetic data with the training clients. INCFL improves on both IPR and preferred-model test accuracy for both smaller (40%) and larger (80%) training data ratios where the IPR improvement of INCFL is larger for the smaller training data ratio.

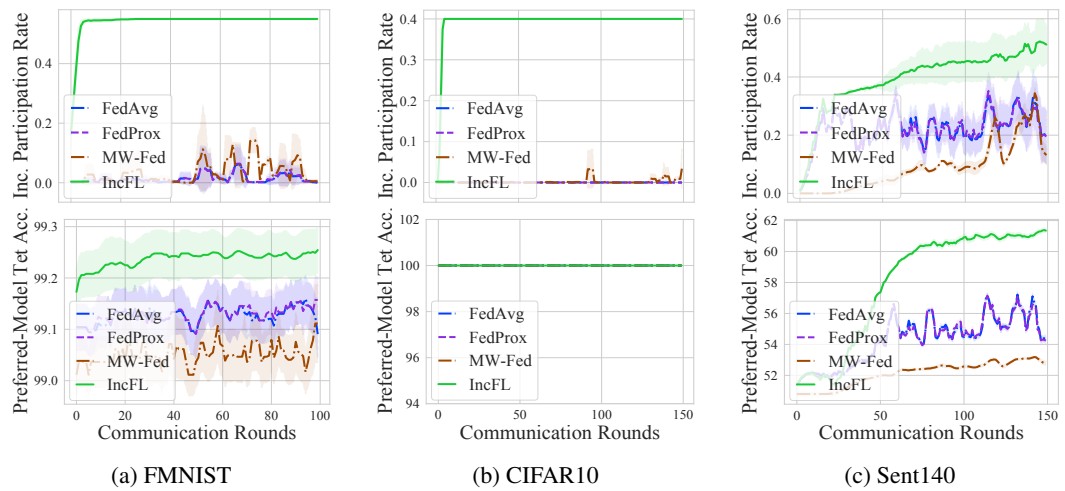

(a) FMNIST     (b) CIFAR10     (c) Sent140

Figure 3: Incentivized participation rate (*upper*), i.e., the fraction of clients incentivized to participate in FL, and preferred-model test accuracy (*lower*) for the training clients' test data with different datasets. For all datasets, INCFL achieves at least 30% to up to 55% increase in the fraction of incentivized clients, while also achieving the maximum preferred-model test accuracy.

standard FedAvg [1], FedProx [2] which aims to tackle data heterogeneity, PerFedAvg [20] which facilitates personalization, and MW-Fed [21] which incentivizes client participation. We provide results on personalization jointly used with INCFL in Appendix F.2.

**Setup.** For Sent140, we consider 308 clients and for the other datasets we have 100 clients that are used for training in FL. These clients are active at some point in training the global model, and we name them as 'seen clients'. In all experiments, 10 clients are sampled every communication round. For FMNIST, data is partitioned into 5 clusters where 2 labels are assigned for each cluster with no labels overlapping across clusters. Clients are randomly assigned to each cluster, and within each cluster, clients are homogeneously distributed with the assigned labels . We similarly partition CIFAR10, where clients are partitioned into 10 clusters with 1 label assigned to each cluster. For the Sent140 dataset, clients are naturally partitioned with their twitter IDs. We also generate 'unseen clients' the same way we generate the seen clients, with 619 clients for Sent140 and 100 clients for all other datasets. These unseen clients represent new incoming clients that have not been seen before during the training rounds of FL. The data of each client is partitioned to $60\% : 40\%$ for training and test data ratio unless mentioned otherwise. Further details are deferred to Appendix F.1.

**Evaluation Metrics: IPR and Preferred-model Test Accuracy.** We evaluate INCFL and other methods with these two key metrics: 1) Incentivized Participation Rate (IPR), defined in (1) and 2) Preferred-Model Test Accuracy. Recall that IPR is the fraction of clients incentivized to participate in FL and use the global server. The preferred-model test accuracy is the average of the clients' test

Table 1: Incentivized participation rate (IPR) and preferred-model test accuracy of the final global models trained with different algorithms for the unseen clients' test data.

| | Incentivized Participation Rate (IPR) | | | Preferred-Model Test Acc. | | |
| | FMNIST | CIFAR10 | Sent140 | FMNIST | CIFAR10 | Sent140 |
|---|---|---|---|---|---|---|
| FedAvg | 0.08 ($\pm$0.01) | 0.00 ($\pm$0.00) | 0.37 ($\pm$0.07) | 98.53 ($\pm$0.13) | 100.00 ($\pm$0.00) | 57.05 ($\pm$1.44) |
| FedProx | 0.07 ($\pm$0.01) | 0.00 ($\pm$0.00) | 0.37 ($\pm$0.07) | 98.43 ($\pm$0.21) | 100.00 ($\pm$0.00) | 57.07 ($\pm$1.42) |
| MW-Fed | 0.05 ($\pm$0.04) | 0.02 ($\pm$0.02) | 0.17 ($\pm$0.03) | 98.32 ($\pm$0.13) | 100.00 ($\pm$0.00) | 55.57 ($\pm$1.28) |
| INCFL | **0.55** ($\pm$0.00) | **0.40** ($\pm$0.00) | **0.43** ($\pm$0.05) | **98.83** ($\pm$0.06) | 100.00 ($\pm$0.00) | **57.16** ($\pm$1.35) |

accuracies computed on their preferred model, either the global model or their solo-trained local model. Higher IPR is beneficial to the server, and higher preferred-model test accuracy is beneficial to the clients. Thus, it is desirable for an algorithm to improve both these metrics.

**Incentivizing the Participation of the Seen Clients.** We first discuss the performance for seen clients used during the training of the global model. In Fig. 2, we show that for the synthetic data, INCFL incentivizes at least 5% more clients compared to the other baselines. The preferred-model test accuracy achieved by INCFL is also highest amongst other baselines. Hence INCFL provides a win-win for both the server and clients since clients have the highest accuracy from choosing the better model and the server has the highest fraction of incentivized clients. In Fig. 3, for all DNN experiments, INCFL significantly improves the IPR to at least 30% to at most 55%. Fig. 3 also shows that the baselines can fail in incentivizing the clients with even 0% clients incentivized. INCFL also improves on the preferred-model test accuracy than the other baselines for FMNIST and Sent140, corroborating the win-win for both the server and clients. For CIFAR10, the preferred-model test accuracy is 100% for all baselines while the IPR is significantly higher for INCFL. This shows that while clients can always achieve 100% by either choosing the local or global model for best performance, server can only gain a large fraction of incentivized clients when using INCFL.

**Incentivizing the Participation of the Unseen Clients.** We now show that INCFL is also bettter at incentivizing the unseen clients that were not active during the training of the global model such as new incoming clients. In Table 1, we show that INCFL consistently improves the IPR of such clients by at least 40% for FMNIST and CIFAR 10, and 6% for Sent140. INCFL also achieves higher or at least the same preferred-model test accuracy compared to that of all baselines for all datasets.

**Effect of Training Data Ratio.** In Fig. 2, we show the performance of INCFL with different ratios of the training data to test data split for each client's data. One can expect that if a client has a high training data ratio, the solo-trained local model of a client sufficiently generalizes well to its test data, and hence the client will be less incentivized to participate in FL. We show in that even if a client has a high training data ratio (80% in Fig. 2(b)), INCFL is able to increase the fraction of incentivized clients compared to other baselines but the improvement is smaller compared to when clients have smaller training data ratio (40% in Fig. 2(a)). In general, clients are believed to have very few labeled training data [22, 23], in which case INCFL can improve the fraction of incentivized clients greatly.

## 5 Concluding Remarks

In this work we carefully re-examine the fundamental assumption in FL that clients always stand to benefit from federation. To do so, we formalize a intuitive notion of client incentives in FL based on whether a global model has better generalization performance than a client's local model. We introduce a novel metric termed as Incentivized Participation Rate (IPR) to explicitly measure the fraction of incentivized clients in FL and develop a corresponding framework INCFL to maximize IPR. In contrast to existing work, INCFL allows the server to play an *active* role in incentivizing clients by dynamically adjusting its aggregation procedure while training the global model. Moreover INCFL is well-suited to both cross-device and cross-silo FL since it stateless and allows partial client availability while training. We provide convergence guarantees for INCFL and show that in practice it can dramatically improve IPR compared to standard FL. We believe our work will open up new research directions in understanding the role played by the server in incentivizing clients for FL. Future work includes jointly examining client incentives with privacy guarantees offered in FL.

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

# A Related Work

**Game-Theoretic Study of Client Coalitions.** Similar to our work, [24, 25] consider the problem of client incentives where each client can either train a local model in isolation or join a coalition with other similar clients to train a common model. The authors study the ratio between the stable solution (where no client is incentivized to shift to a different coalition) and the optimal solution (where the sum of losses of all clients is minimized). Although this line of work establishes a number of useful game-theoretic insights, it focuses on simple mean estimation and linear regression problems. The server plays the role of a passive matchmaker, facilitating the formation of coalitions of clients. In contrast, in our work, the server actively seeks to train a single global model that maximizes client participation. This perspective alleviates some of the analysis complexities occurring in game-theoretic formulations and allows us to consider general non-convex objective functions.

**Client Incentives for Contributing Data and Computation Resources.** Other recent works such as [21, 26–28] develop mechanisms to incentivize clients to contribute data samples and local computational resources to federated training, and compensate for these contributions. In [21], which is closest to our work, the authors consider linear problems and analyze the existence and optimality of equilibria in the problem of equitably distributing the burden of data contribution between clients. They propose a heuristic algorithm called Multiplicative Weight (MW)-Fed, where the server instructs clients for whom the global model is performing poorly to conduct more local updates. Unlike our work, this approach does not explicitly maximize client participation, and it is not supported by theoretical guarantees. However, we include it as a baseline in our experiments (Section 4).

**Personalized and Fair Federated Learning.** Finally, personalized or fair FL methods may offer an auxiliary benefit of increasing incentivized client participation even though they are not formally studied in the client incentives context. Instead of having all clients use a common global model, personalized FL methods consider learning models unique to each client. In cross-device settings, a common approach is to consider methods that fine-tune the global model to produce personalized models [29, 30, 7, 31, 32]. This can naturally incentivize more clients to use the global model and participate in FL. Our INCFL algorithm is orthogonal to and can be combined with such personalized FL methods. In our experiments, we demonstrate the performance of INCFL combined with local fine-tuning and compare it with [20], which uses meta-learning for personalization. Another related area is fair FL, where a common goal is to train a global model whose accuracy has less variance across the client population than standard FedAvg [33, 34]. A side benefit of these methods is that they can incentivize the worst performing clients to participate. However, a downside is that the performance of the global model may be degraded for the best performing clients, thus incentivizing them to leave the federation. We show in additional experimental results in Appendix F.1 that common fair FL methods are indeed not effective in improving the overall client participation rate.

# B Mean Estimation as a Toy Example for INCFL

## B.1 Maximizing IPR in Two Client Mean Estimation

We consider a setup with $M = 2$ clients where each client aims to find the mean of its data distribution by minimizing the true loss function $f_k(w) = \mathbb{E}_{\xi_k}\left[(w - \xi_k)^2\right]$, $\xi_k \sim \mathcal{N}(\theta_k, \nu^2)$ $\forall k \in [2]$. In practice, clients only have $N_k$ samples drawn from their distribution denoted by $\mathcal{B}_k = \{e_{k,j}\}_{j=1}^{N_k}$ and can only minimize their empirical loss function given by $F_k(w) = \frac{1}{|\mathcal{B}_k|}\sum_{j=1}^{N_k}(w - e_{k,j})^2$. Then the solo trained models at each client will be their local empirical mean, i.e. $\widehat{w}_k = \widehat{\theta}_k = \frac{1}{|\mathcal{B}_k|}\sum_{j=1}^{N_k} e_{k,j}$.

**IPR for Standard FL Model Decreases Exponentially with Heterogeneity.** For simplicity let us assume $N_1 = N_2 = N$. Let $\gamma^2 = \nu^2/N$ be the variance of the local empirical means and $\gamma_G^2 = ((\theta_1 - \theta_2)/2)^2 > 0$ be a measure of heterogeneity between the true means. The standard FL objective will always set the FL model to be the average of the local empirical means (i.e. $w = (\widehat{\theta}_1 + \widehat{\theta}_2)/2$) and does not take into account the heterogeneity among the clients. As a result, the IPR of the global model decreases *exponentially* as $\gamma_G^2$ increases.

**Lemma B.1.** *The expected IPR of the standard FL model is upper bounded by* $2\exp\left(-\frac{\gamma_G^2}{5\gamma^2}\right)$, *where the expectation is taken over the randomness in the local datasets* $\mathcal{B}_1, \mathcal{B}_2$.

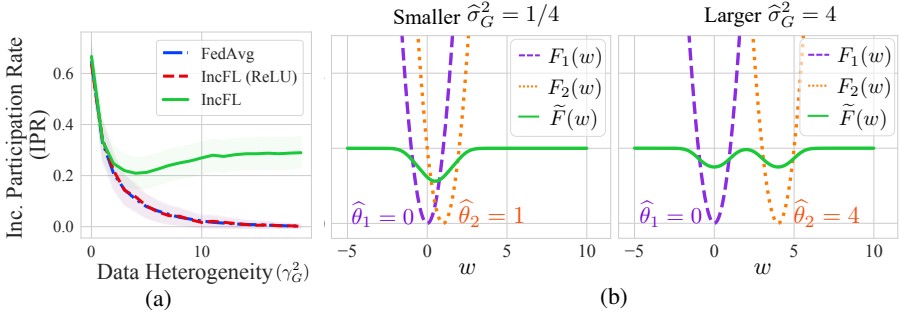

Figure 4: Results for the two client mean estimation in Appendix B.1; (a): IPR for FedAvg decays exponentially while IPR for INCFL is lower bounded by a constant. Replacing the sigmoid approximation with ReLU approximation in INCFL leads to the same solution as FedAvg; (b): IncFL adapts to the heterogeneity of the problem—for small heterogeneity it encourages collaboration by having a single global minima, for large heterogeneity it encourages separation by having far away local minimas.

**Maximizing IPR with Relaxed Objective.** We now explicitly maximize the IPR for this setting by solving for a relaxed version of the objective in (2) as proposed earlier. We replace the true loss $f_k(\cdot)$ by the empirical loss $F_k(\cdot)$ and replace the 0-1 (sign) loss with a differentiable approximation $h(\cdot)$.

We first show that setting $h(\cdot)$ to be a standard convex surrogate for the 0-1 loss (e.g. log loss, exponential loss, ReLU) leads to our new objective behaving the same as the standard FL objective.

**Lemma B.2.** *Let $h$ be any function that is convex, twice differentiable, and strictly increasing in $[0, \infty)$. Then our relaxed objective is strictly convex and has a unique minimizer at $w^* = \left(\frac{\widehat{\theta}_1 + \widehat{\theta}_2}{2}\right)$.*

**Maximizing the INCFL Objective Leads to Increased IPR.** Based on Lemma B.2, we see that we need nonconvexity in $h(\cdot)$ for the objective to behave differently than standard FL. We set $h(x) = \sigma(x) = \frac{\exp(x)}{1+\exp(x)}$, as proposed in our INCFL objective in (3). We find that the INCFL objective *adapts* to the empirical heterogeneity parameter $\widehat{\gamma}_G^2 = \left(\frac{\widehat{\theta}_2 - \widehat{\theta}_1}{2}\right)^2$. If $\widehat{\gamma}_G^2 < 1$ (small data heterogeneity), the objective encourages collaboration by setting the global model to be the average of the local models. On the other hand, if $\widehat{\gamma}_G^2 > 2$ (large data heterogeneity), the objective encourages *separation* by setting the global model close to either the local model of the first client or the local model of the second client (see Fig. 4). Based on this observation, we have the following theorem.

**Theorem B.1.** *Let $w$ be a local minima of the INCFL objective. The expected IPR using $w$ is lower bounded by $\frac{1}{16}\exp\left(-\frac{1}{\gamma^2}\right)$ where the expectation is over the randomness in the local dataset $\mathcal{B}_1, \mathcal{B}_2$.*

Note that our result above is independent of the heterogeneity parameter $\gamma_G^2$. Therefore even with $\gamma_G^2 \gg 0$, INCFL will keep incentivizing atleast one client by adapting its objective accordingly. Additional discussion and proof details can be found in Appendix B.

We begin by recalling the setup discussed in Appendix B.1. We have a setup with $M = 2$ clients where each client aims to find the mean of its data distribution by minimizing the true loss function $f_k(w) = \mathbb{E}_{\xi_k}\left[(w - \xi_k)^2\right], \xi_k \sim \mathcal{N}(\theta_k, \nu^2) \ \forall k \in [2]$. Without loss of generality we assume that $\theta_2 \geq \theta_1$. In practice, each client has $N$ samples drawn from their distribution denoted by $\mathcal{B}_k = \{e_{k,j}\}_{j=1}^N$ and can only minimize their empirical loss function given by

$$F_k(w) = \frac{1}{N}\sum_{j=1}^N (w - e_{k,j})^2 \tag{8}$$

$$= (w - \widehat{\theta}_k)^2 + \frac{1}{N}\sum_{j=1}^N (\widehat{\theta}_k - e_{k,j})^2 \tag{9}$$

where $\widehat{\theta}_k = \frac{1}{N}\sum_{j=1}^N e_{k,j}$ is the empirical mean at client $k$. We assume that clients set their solo trained models as their empirical mean, i.e. $\widehat{w}_k = \widehat{\theta}_k$.

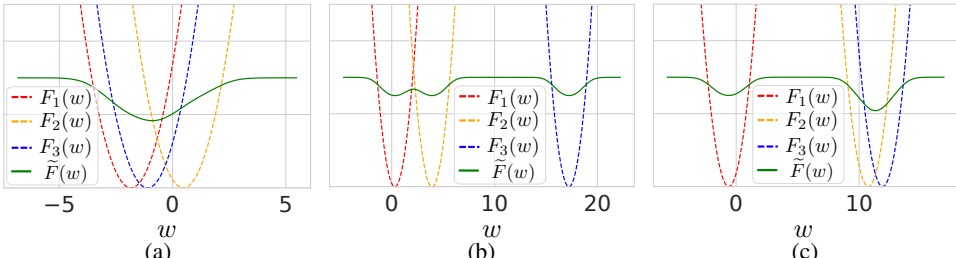

Figure 5: Results for the three client mean estimation; (a): case 1 when the true mean across clients are close to amongst each other where IncFL's optimal soluation is identical to that of FedAvg; (b): case 2 when the true mean across clients are all different from each other where IncFL's optimal solution ensures that at least one of the clients will be incentivized participate with IncFL's global model (unlike FedAvg); (c) case 3 when two clients' true means are close to each other while the other client has a different mean. IncFL in this case, is able to ensure that the two clients participate while FedAvg is not able to make any client participate.

We define the following quantities

$$\gamma^2 := \frac{\nu^2}{N}; \quad \gamma_G^2 = \left(\frac{\theta_2 - \theta_1}{2}\right)^2; \quad (10)$$

Note that the distribution of the empirical means itself follows a normal distribution following the linear additivity of independent normal random variables.

$$\widehat{\theta}_1 \sim \mathcal{N}(\theta_1, \gamma^2); \quad \widehat{\theta}_2 \sim \mathcal{N}(\theta_2, \gamma^2) \quad (11)$$

### B.2 Maximizing IPR in Three Client Mean Estimation.

We further examine the property of IncFL to incentivize clients with a 3 clients toy example which is an extension from what we have shown in Appendix B.1 for 2 clients. Reusing the notation from Appendix B.1, where $\theta_i$ is the true mean at client $i$ and $\hat{\theta}_i \sim \mathcal{N}(\theta_i, 1)$ is the empirical mean of a client, our analysis can be divided into the following cases for the 3 client example (see Fig. 5):

- Case 1: $\theta_1 \approx \theta_2 \approx \theta_3$: This case captures the setting where the data at the clients is almost i.i.d. In this case, it makes sense for clients to collaborate together and therefore IncFL's optimal solution will be the average of local empirical means (same as FedAvg).

- Case 2: $\theta_1 \neq \theta_2 \neq \theta_3$: This case captures the setting where the data at clients is completely disparate. In this case, none of the clients benefit from collaborating and therefore IncFL's optimal solution will be the local model of one of the clients. This ensures atleast one of the clients will still be incentivized to use the IncFl global model unlike FedAvg.

- Case 3: $\theta_1 \approx \theta_2 \neq \theta_3$: The most interesting case happens when data at two of the clients is similar but the data at the third client is different. Without loss of generality we assume that data at clients 1 and 2 is similar and client 3 is different. In this case, although client 1 and 2 benefit from federating, FedAvg is unable to leverage that due to the heterogeneity at client 3. IncFL, on the other hand, will set the optimal solution to be the average of the local models of just client 1 and client 2. This ensures that clients 1 and 2 will both continue to participate in the FL training process, thus maximizing the number of incentivized clients.

The behavior of IncFL in the three client setup clearly highlights the non-trivialness of our proposed IncFL's formulation.

### B.3 Proof of Lemma B.1

**Lemma B.1** *The expected IPR of the standard FL model is upper bounded by* $2\exp\left(-\frac{\gamma_G^2}{5\gamma^2}\right)$, *where the expectation is taken over the randomness in the local datasets* $\mathcal{B}_1, \mathcal{B}_2$.

**Proof.**

The standard FL model is given by,

$$w = \frac{\widehat{\theta}_1 + \widehat{\theta}_2}{2} \tag{12}$$

Therefore the expected IPR is,

$$\mathbb{E}\left[\frac{\mathbb{I}\{(w-\theta_1)^2 < (\widehat{\theta}_1 - \theta_1)^2\} + \mathbb{I}\{(w-\theta_2)^2 < (\widehat{\theta}_2 - \theta_2)^2\}}{2}\right] \tag{13}$$

$$= \frac{1}{2}\left[\underbrace{\mathbb{P}\left((w-\theta_1)^2 < (\widehat{\theta}_1 - \theta_1)^2\right)}_{T_1} + \underbrace{\mathbb{P}\left((w-\theta_2)^2 < (\widehat{\theta}_2 - \theta_2)^2\right)}_{T_2}\right] \tag{14}$$

Next we bound $T_1$ and $T_2$. Bounding $T_1$ :

$$T_1 = \mathbb{P}\left((w-\theta_1)^2 < (\widehat{\theta}_1 - \theta_1)^2\right) \tag{15}$$

$$= \mathbb{P}\left(\left(\frac{\widehat{\theta}_1 + \widehat{\theta}_2}{2} - \theta_1\right)^2 < (\widehat{\theta}_1 - \theta_1)^2\right) \tag{16}$$

$$= \mathbb{P}\left(\left(\frac{\widehat{\theta}_2 - \widehat{\theta}_1}{2}\right)^2 + 2\left(\frac{\widehat{\theta}_2 - \widehat{\theta}_1}{2}\right)(\widehat{\theta}_1 - \theta_1) < 0\right) \tag{17}$$

$$= \mathbb{P}\left(\left\{\left(\frac{\widehat{\theta}_2 - \widehat{\theta}_1}{2}\right)^2 + 2\left(\frac{\widehat{\theta}_2 - \widehat{\theta}_1}{2}\right)(\widehat{\theta}_1 - \theta_1) < 0\right\} \cap \left\{\widehat{\theta}_2 > \widehat{\theta}_1\right\}\right)$$
$$+ \mathbb{P}\left(\left\{\left(\frac{\widehat{\theta}_2 - \widehat{\theta}_1}{2}\right)^2 + 2\left(\frac{\widehat{\theta}_2 - \widehat{\theta}_1}{2}\right)(\widehat{\theta}_1 - \theta_1) < 0\right\} \cap \left\{\widehat{\theta}_2 \le \widehat{\theta}_1\right\}\right) \tag{18}$$

$$= \mathbb{P}\left(\left\{\left(\frac{\widehat{\theta}_2 - \widehat{\theta}_1}{2}\right) + 2(\widehat{\theta}_1 - \theta_1) < 0\right\} \cap \left\{\widehat{\theta}_2 > \widehat{\theta}_1\right\}\right)$$
$$+ \mathbb{P}\left(\left\{\left(\frac{\widehat{\theta}_2 - \widehat{\theta}_1}{2}\right)^2 + 2\left(\frac{\widehat{\theta}_2 - \widehat{\theta}_1}{2}\right)(\widehat{\theta}_1 - \theta_1) < 0\right\} \cap \left\{\widehat{\theta}_2 \le \widehat{\theta}_1\right\}\right) \tag{19}$$

$$\le \mathbb{P}\left(\left(\frac{\widehat{\theta}_2 - \widehat{\theta}_1}{2}\right) + 2(\widehat{\theta}_1 - \theta_1) < 0\right) + \mathbb{P}\left(\widehat{\theta}_2 - \widehat{\theta}_1 \le 0\right) \tag{20}$$

$$= \mathbb{P}\left(Z_1 < 0\right) + \mathbb{P}\left(Z_2 \le 0\right) \quad \text{where } Z_1 \sim \mathcal{N}\left(\gamma_G, \frac{5}{2}\gamma^2\right), Z_2 \sim \mathcal{N}\left(2\gamma_G, 2\gamma^2\right) \tag{21}$$

$$\le \exp\left(-\frac{\gamma_G^2}{5\gamma^2}\right) + \exp\left(-\frac{\gamma_G^2}{\gamma^2}\right) \tag{22}$$

$$\le 2\exp\left(-\frac{\gamma_G^2}{5\gamma^2}\right) \tag{23}$$

where (18) uses $\mathbb{P}(A) = \mathbb{P}(A \cap B) + \mathbb{P}\left(A \cap B^{\complement}\right)$, (20) uses $\mathbb{P}(A \cap B) \le \mathbb{P}(A)$, (21) uses (11) and linear additivity of independent normal random variables, (22) uses a Chernoff bound.

We can similarly bound $T_2$ to get $T_2 \le 2\exp\left(-\frac{\gamma_G^2}{5\gamma^2}\right)$. Thus the expected IPR of the standard FL model is upper bounded by $2\exp\left(-\frac{\gamma_G^2}{5\gamma^2}\right)$.

### B.4  Proof of Lemma B.2

**Lemma B.2** *Let $h$ be any function that is convex, twice differentiable, and strictly increasing in $[0, \infty)$. Then our relaxed objective is strictly convex and has a unique minimizer at $w^* = \left( \frac{\widehat{\theta}_1 + \widehat{\theta}_2}{2} \right)$.*

**Proof.**

Let us denote our relaxed objective by $v(w)$. Then $v(w)$ can be written as,

$$v(w) = \frac{1}{2} \left[ h \left( F_1(w) - F(\widehat{w}_1) \right) + h \left( F_2(w) - F(\widehat{w}_2) \right) \right] \tag{24}$$

$$= \underbrace{\frac{1}{2} h \left( (w - \widehat{\theta}_1)^2 \right)}_{v_1(w)} + \underbrace{\frac{1}{2} h \left( (w - \widehat{\theta}_2)^2 \right)}_{v_2(w)} \tag{25}$$

$$\tag{26}$$

We first prove that $v_1(w)$ is strictly convex. Let $\lambda \in (0, 1)$ and $(w_1, w_2)$ be any pair of points in $\mathbb{R}^2$ such that $w_1 \neq w_2$. We have,

$$v_1(\lambda w_1 + (1 - \lambda) w_2) = \frac{1}{2} h \left( (\lambda(w_1 - \widehat{\theta}_1) + (1 - \lambda)(w_2 - \widehat{\theta}_1))^2 \right) \tag{27}$$

$$< \frac{1}{2} h \left( \lambda(w_1 - \widehat{\theta}_1)^2 + (1 - \lambda)(w_2 - \widehat{\theta}_1)^2 \right) \tag{28}$$

$$\leq \frac{\lambda}{2} h \left( (w_1 - \widehat{\theta}_1)^2 \right) + \frac{1 - \lambda}{2} h \left( (w_2 - \widehat{\theta}_1)^2 \right) \tag{29}$$

$$= \lambda v_1(w_1) + (1 - \lambda) v_1(w_2) \tag{30}$$

where (28) follows from the strict convexity of $f(w) = w^2$ and the fact that $h(w)$ is strictly increasing in the range $[0, \infty)$, (29) follows from the convexity of $h(w)$.

This completes the proof that $v_1(w)$ is strictly convex. We can similarly prove that $v_2(w)$ is stricly convex and hence $v(w)$ is strictly convex since summation of strictly convex functions is strictly convex.

Also note that,

$$\nabla v(w) = \nabla h \left( (w - \widehat{\theta}_1)^2 \right) (w - \widehat{\theta}_1) + \nabla h \left( (w - \widehat{\theta}_2)^2 \right) (w - \widehat{\theta}_2) \tag{31}$$

It is easy to see that $\nabla v(w) = 0$ at $w = \left( \frac{\widehat{\theta}_1 + \widehat{\theta}_2}{2} \right)$. Since $v(w)$ is strictly convex this implies that $w^* = \left( \frac{\widehat{\theta}_1 + \widehat{\theta}_2}{2} \right)$ will be a unique global minimizer. This completes the proof.

### B.5  Proof of Theorem B.1

Before stating the proof of Theorem 3.1 we first state some intermediate results that will be used in the proof.

The INCFL objective can be written as,

$$v(w) = \frac{1}{2} \sigma \left( (w - \widehat{\theta}_1)^2 \right) + \frac{1}{2} \sigma \left( (w - \widehat{\theta}_2)^2 \right) \tag{32}$$

where $\sigma(w) = 1/(1 + \exp(-w))$.

We additionally define the following quantities,

$$i := \operatorname{argmin} \left\{ \widehat{\theta}_1, \widehat{\theta}_2 \right\}; \quad j := \operatorname{argmax} \left\{ \widehat{\theta}_1, \widehat{\theta}_2 \right\}; \quad \widehat{\gamma}_G := \frac{\widehat{\theta}_j - \widehat{\theta}_i}{2} \tag{33}$$

Let $q(w) = \sigma(w)(1 - \sigma(w))$. The gradient of $v(w)$ is given as,

$$\nabla v(w) = q \left( (w - \widehat{\theta}_1)^2 \right) (w - \widehat{\theta}_1) + q \left( (w - \widehat{\theta}_2)^2 \right) (w - \widehat{\theta}_2) \tag{34}$$

**Lemma B.3** *For $\widehat{\gamma}_G > 2$, $w = \left(\frac{\widehat{\theta}_1 + \widehat{\theta}_2}{2}\right)$ will be a local maxima of the* INCFL *objective.*

It is easy to see that $w = \left(\frac{\widehat{\theta}_1 + \widehat{\theta}_2}{2}\right)$ will always be a stationary point of $\nabla v(w)$. Our goal is to determine whether it will be a local minima or a local maxima. To do so, we calculate the hessian of $v(w)$ as follows. Let $f(w) = 2\sigma(w)(1 - \sigma(w))(1 - 2\sigma(w))$. Then,

$$\nabla^2 v(w) = \underbrace{f\left((w - \widehat{\theta}_1)^2\right)(w - \widehat{\theta}_1)^2 + q\left((w - \widehat{\theta}_1)^2\right)}_{h_1(w)} + \underbrace{f\left((w - \widehat{\theta}_2)^2\right)(w - \widehat{\theta}_2)^2 + q\left((w - \widehat{\theta}_2)^2\right)}_{h_2(w)}$$

(35)

Note that $h_1(w) = h_2(w)$ for $w = \left(\frac{\widehat{\theta}_1 + \widehat{\theta}_2}{2}\right)$. Hence it suffices to focus on the condition for which $h_1(w) < 0$ at $w = \left(\frac{\widehat{\theta}_1 + \widehat{\theta}_2}{2}\right)$. We have,

$$h_1\left((\widehat{\theta}_1 + \widehat{\theta}_2)/2\right) = f(\widehat{\gamma}_G^2)\widehat{\gamma}_G^2 + q(\widehat{\gamma}_G^2) \tag{36}$$

$$= q(\widehat{\gamma}_G^2)(2(1 - 2\sigma(\widehat{\gamma}_G^2))\widehat{\gamma}_G^2 + 1) \tag{37}$$

$$< 0 \quad \text{for } \widehat{\gamma}_G \geq 1.022 \tag{38}$$

where the last inequality follows from the fact that $q(w) > 0$ for all $w \in \mathbb{R}$ and $2(1 - 2\sigma(w^2))w^2 + 1 < 0$ for $w \geq 1.022$. Thus for $\widehat{\gamma}_G > 2$, $w = \left(\frac{\widehat{\theta}_1 + \widehat{\theta}_2}{2}\right)$ will be a local maxima of the INCFL objective.

**Lemma B.4** *For $\widehat{\gamma}_G > 0$, any local minima of $v(w)$ lies in the range $(\widehat{\theta}_i, \widehat{\theta}_i + 2] \cup [\widehat{\theta}_j - 2, \widehat{\theta}_j)$.*

Firstly note that since $\widehat{\gamma}_G > 0$ we have $\widehat{\theta}_j > \widehat{\theta}_i$. Secondly note that since $q(w) > 0$ for all $w \in \mathbb{R}$, $\nabla v(w) < 0$ for all $w \leq \widehat{\theta}_i$ and $\nabla v(w) > 0$ for all $w \geq \widehat{\theta}_j$. Therefore any root of the function $\nabla v(w)$ must lie in the range $(\widehat{\theta}_i, \widehat{\theta}_j)$.

**Case 1:** $0 < \widehat{\gamma}_G \leq 2$.

In this case, the lemma is trivially satisified since $(\widehat{\theta}_i, \widehat{\theta}_j) \subset \left\{(\widehat{\theta}_i, \widehat{\theta}_i + 2] \cup [\widehat{\theta}_j - 2, \widehat{\theta}_j)\right\}$.

**Case 2:** $\widehat{\gamma}_G > 2$.

Let $x = w - \widehat{\theta}_i$ and $g(x) = q(x^2)x$. We can write $\nabla v(w)$ as,

$$\nabla v(\widehat{\theta}_i + x) = g(x) - g(2\widehat{\gamma}_G - x) \tag{39}$$

It can be seen that for $x > 2$, $g(x)$ is a decreasing function. For $x \in (2, \widehat{\gamma}_G)$ we have $x > 2\widehat{\gamma}_G - x$ which implies $g(x) > g(2\widehat{\gamma}_G - x)$. Therefore $\nabla v(\widehat{\theta}_i + x) > 0$ for $x \in (2, \widehat{\gamma}_G)$. Also $\nabla v(\widehat{\theta}_i + 2\widehat{\gamma}_G - x) = -\nabla v(\widehat{\theta}_i + x)$ and therefore $\nabla v(\widehat{\theta}_i + x) < 0$ for $x \in (\widehat{\gamma}_G, 2\widehat{\gamma}_G - 2)$. $\nabla v(\widehat{\theta}_i + \widehat{\gamma}_G) = 0$ but this will be a local maxima for $\widehat{\gamma}_G > 2$ as shown in Lemma B.3. Thus there exists no local minima of $v(w)$ for $w \in (\widehat{\theta}_i + 2, \widehat{\theta}_j - 2)$

Combining both cases we see that any local minima of $v(w)$ lies in the range $\left\{(\widehat{\theta}_i, \widehat{\theta}_i + 2] \cup [\widehat{\theta}_j - 2, \widehat{\theta}_j)\right\}$.

**Theorem B.1** *Let $w$ be a local minima of the* INCFL *objective. The expected IPR using $w$ is lower bounded by $\frac{1}{16}\exp\left(-\frac{1}{\gamma^2}\right)$ where the expectation is over the randomness in the local dataset $\mathcal{B}_1, \mathcal{B}_2$.*

**Proof.**

The IPR can be written as,

$$\frac{1}{2}\left[\mathbb{P}\left((w-\theta_i)^2 < (\widehat{\theta}_i - \theta_i)^2\right) + \mathbb{P}\left((w-\theta_j)^2 < (\widehat{\theta}_j - \theta_j)^2\right)\right] \tag{40}$$

We focus on the case where $\widehat{\theta}_2 \neq \widehat{\theta}_i$ implying $\widehat{\theta}_j > \widehat{\theta}_i$ ($\widehat{\theta}_2 = \widehat{\theta}_1$ is a zero-probability event and does not affect our proof). Let $w$ be any local minima of the INCFL objective. From Lemma B.4 we know that $w$ will lie in the range $(\widehat{\theta}_i, \widehat{\theta}_i + 2] \cup [\widehat{\theta}_j - 2, \widehat{\theta}_j)$

**Case 1:** $w \in (\widehat{\theta}_i, \widehat{\theta}_i + 2]$

$$\mathbb{P}\left((w-\theta_i)^2 < (\widehat{\theta}_i - \theta_i)^2\right) = \mathbb{P}\left((w-\widehat{\theta}_i)^2 + 2(w-\widehat{\theta}_i)(\widehat{\theta}_i - \theta_i) < 0\right) \tag{41}$$

$$= \mathbb{P}\left((w-\widehat{\theta}_i) + 2(\widehat{\theta}_i - \theta_i) < 0\right) \tag{42}$$

$$\geq \mathbb{P}\left(2 + 2(\widehat{\theta}_i - \theta_i) < 0\right) \tag{43}$$

$$= \mathbb{P}\left((\widehat{\theta}_i - \theta_i) < -1\right) \tag{44}$$

$$\geq \mathbb{P}\left(\left\{\widehat{\theta}_1 < \widehat{\theta}_2\right\} \cap \left\{(\widehat{\theta}_1 - \theta_1) < -1\right\}\right) \tag{45}$$

$$= \mathbb{P}\left(\widehat{\theta}_1 < \widehat{\theta}_2\right)\mathbb{P}\left(\widehat{\theta}_1 - \theta_1 < -1 | \widehat{\theta}_1 < \widehat{\theta}_2\right) \tag{46}$$

$$\geq \mathbb{P}\left(\widehat{\theta}_1 < \widehat{\theta}_2\right)\mathbb{P}\left(\widehat{\theta}_1 - \theta_1 < -1\right) \tag{47}$$

$$= \mathbb{P}\left(\widehat{\theta}_1 < \widehat{\theta}_2\right)\mathbb{P}\left(Z > 1/\gamma\right) \quad \text{where } Z \sim \mathcal{N}(0,1) \tag{48}$$

$$\geq \frac{1}{8}\exp\left(-\frac{1}{\gamma^2}\right) \tag{49}$$

(42) uses the fact that $(w-\widehat{\theta}_i) > 0$, (43) uses $(w-\widehat{\theta}_i) \leq 2$, (45) uses $\mathbb{P}(A) \geq \mathbb{P}(A \cap B)$ and definition of $i$. (47) uses the following argument. If $\theta_1 - 1 \geq \widehat{\theta}_2$ then $\mathbb{P}\left(\widehat{\theta}_1 - \theta_1 < -1 | \widehat{\theta}_1 < \widehat{\theta}_2\right) = 1$. If $\theta_1 - 1 < \widehat{\theta}_2$ then $\mathbb{P}\left(\widehat{\theta}_1 - \theta_1 < -1 | \widehat{\theta}_1 < \widehat{\theta}_2\right) = \mathbb{P}\left(\widehat{\theta}_1 - \theta_1 < -1\right)/\mathbb{P}\left(\widehat{\theta}_1 < \widehat{\theta}_2\right) \geq \mathbb{P}\left(\widehat{\theta}_1 - \theta_1 < -1\right)$. (48) uses $\widehat{\theta}_1 - \theta_1 \sim \mathcal{N}(0, \gamma^2)$, (49) uses $\mathbb{P}\left(\widehat{\theta}_1 < \widehat{\theta}_2\right) \geq \frac{1}{2}$ and $\mathbb{P}(Z \geq x) \geq \frac{2\exp(-x^2/2)}{\sqrt{2\pi}(\sqrt{4+x^2}+x)} \geq \frac{1}{4}\exp(-x^2)$ where $Z \sim \mathcal{N}(0,1)$ [35].

In the case where $w \in (\widehat{\theta}_j - 2, \widehat{\theta}_j]$ a similar technique can be used to lower bound $\mathbb{P}\left((w-\theta_j)^2 < (\widehat{\theta}_j - \theta_j)^2\right)$. Thus the IPR of any local minima of the INCFL objective is lower bounded by $\frac{1}{16}\exp\left(-\frac{1}{\gamma^2}\right)$.

## C Additional Discussion on Our Proposed INCFL Algorithm

We first present our pseudo-code for INCFL below in Algorithm 1.

---

**Algorithm 1** Proposed Client Incentivizing FL Framework: INCFL

1: **Input:** mini-batch size $b$, local iteration steps $\tau$, training loss $F_i(\widehat{\mathbf{w}}_i)$ for each client $i \in [M]$
2: **Output:** Global model $\mathbf{w}^{(T,0)}$, **Initialize:** Global model $\mathbf{w}^{(0,0)}$
3: **For** $t = 0, ..., T-1$ **communication rounds do**:
4:     **Global server do:**
5:         Select $m$ clients for $\mathcal{S}^{(t,0)}$ uniformly at random and send $\mathbf{w}^{(t,0)}$ to clients in $\mathcal{S}^{(t,0)}$
6:     **Clients $k \in \mathcal{S}^{(t,0)}$ in parallel do:**
7:         Set $\mathbf{w}_k^{(t,0)} = \mathbf{w}^{(t,0)}$, and calculate $q_k(\mathbf{w}_k^{(t,0)}) = \sigma(F_k(\mathbf{w}_k^{(t,0)}) - F_k(\widehat{\mathbf{w}}_k))$
8:         **For $r = 0, ..., \tau-1$ local iterations do:**
9:           Update $\mathbf{w}_k^{(t,r+1)} \leftarrow \mathbf{w}_k^{(t,r)} - \eta_l \mathbf{g}(\mathbf{w}_k^{(t,r)}, \xi_k^{(t,r)})$
10:        Send $\Delta\mathbf{w}_k^{(t,0)} = \mathbf{w}_k^{(t,0)} - \mathbf{w}_k^{(t,\tau)}$ and aggregation weight $q_k(\mathbf{w}_k^{(t,0)})$ to the server
11:     **Global server do:**
12:         Update global model with $\mathbf{w}^{(t+1,0)} = \mathbf{w}^{(t,0)} - \eta_g^{(t,0)} \sum_{k \in \mathcal{S}^{(t,0)}} q_k(\mathbf{w}^{(t,0)}) \Delta\mathbf{w}_k^{(t,0)}$

---

**Adaptive Server Learning Rate for INCFL.** With $L_c$ continuous and $L_s$ smooth $F_k(\mathbf{w})$, $\forall k \in [M]$ (see Assumption C.1), the objective $\widetilde{F}(\mathbf{w})$ is $\widetilde{L}_s$ smooth where $\widetilde{L}_s = \frac{L_s}{M}\sum_{k=1}^{M} q_k(\mathbf{w}) + \frac{L_c}{4}$ (see Appendix D.2). Hence, the optimal learning rate $\tilde{\eta}$ for the INCFL is given by, $\widetilde{\eta} = 1/\widetilde{L}_s = M\eta/\left(\sum_{k=1}^{M} q_k(\mathbf{w}) + \epsilon\right)$, where $\eta = \frac{1}{L_s}$ is the optimal learning rate for standard FL and $\epsilon = \frac{ML_c}{4L_s}$ > 0 is a constant. The denominator of the optimal $\widetilde{\eta}$ is proportional to the sum of the aggregation weights $q_k(\mathbf{w})$ and acts as a dynamic normalizing factor. Therefore, we propose using adaptive global learning rate $\eta_g^{(t,0)} = \eta_g/(\sum_{k \in \mathcal{S}^{(t,0)}} q_k(\mathbf{w}^{(t,0)}) + \epsilon)$ with hyperparameters $\eta_g$ and $\epsilon$.

**INCFL's Theoretical Learning Rate Behavior for Fig. 4 (b).** Here, we provide a plot of INCFL's theoretical learning rate for the mean estimation example in Fig. 4(b) in Fig. 6 to show how the learning rate changes for different regions of the model. We show this plot as a proof of concept on the adaptive learning rate we discuss above. For the sigmoid function which is used for our INCFL objective, using a global notion of smoothness can cause gradient descent to be too slow since global smoothness is determined by behavior at $w = 0$ where $w$ is the model. In this case, it is better to use a local estimate of smoothness in the flat regions where $|w| >> 0$. Recall that $\nabla^2 \sigma(w) = \sigma(w)(1 - \sigma(w))(1 - 2\sigma(w)) < \sigma(x)(1 - \sigma(w))$ and therefore setting the learning rate proportional to $\frac{1}{\sigma(w)(1-\sigma(w))}$ can increase the learning rate in flat regions where $\sigma(w)$ is close to 1 or 0. Following a similar argument, we can show that the learning rate in our objective should be proportional to $1/\left(\sum_{i=1}^{M} \sigma(F_i(w) - F_i(\hat{w}^*))(1 - \sigma(F_i(w) - F_i(\hat{w}^*)))\right)$.

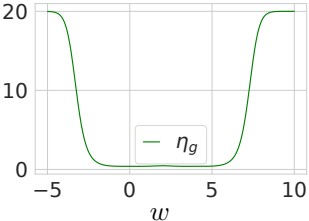

Figure 6: Behavior of Theoretical Learning Rate of IN-CFL for the mean estimation example in Fig. 4(b). As expected from the theoretical learning rate formula, we see a higher learning rate in regions where the function is flat.

**Ease of Implementing INCFL.** INCFL enjoys the following properties: i) it does not modify the local SGD procedure clients perform in standard FL, ii) it allows for partial client participation, and iii) it is stateless. By stateless, we mean that clients do not carry varying local parameters throughout training rounds preventing any problems from stale parameters which can be exacerbated with partical client participation. Note that $F_k(\widehat{\mathbf{w}}_k)$ needed for calculating $q_k(\mathbf{w}), k \in [M]$ can be considered as an input parameter for INCFL, that is computed once and saved as a constant beforehand at each client by training $\widehat{\mathbf{w}}_k$ on its local dataset for a few SGD steps.

## C.1 Convergence Properties of INCFL

In this section we show the convergence properties of the global model trained with INCFL. Our convergence analysis shows that the gradient norm of our global model goes to zero and therefore we converge to a stationary point of our objective $\widetilde{F}(\mathbf{w})$.

First we introduce the assumptions and definitions utilized for our convergence analysis below.

**Assumption C.1** (Continuity & Smoothness of $F_k(\mathbf{w})$, $\forall\,k$)**.** *The local objective functions for all clients, $F_1(\mathbf{w})$, ..., $F_M(\mathbf{w})$, are all $L_c$-continuous and $L_s$-smooth for all $\mathbf{w}$.*

**Assumption C.2** (Unbiased Stochastic Gradient with Bounded Variance for $F_k(\mathbf{w})$, $\forall\,k$)**.** *For the mini-batch $\xi_k$ uniformly sampled at random from $\mathcal{B}_k$ from user $k$, the resulting stochastic gradient is unbiased, i.e., $\mathbb{E}[\mathbf{g}_k(\mathbf{w}_k, \xi_k)] = \nabla F_k(\mathbf{w}_k)$. Also, the variance of stochastic gradients is bounded: $\mathbb{E}[\|\mathbf{g}_k(\mathbf{w}_k, \xi_k) - \nabla F_k(\mathbf{w}_k)\|^2] \le \sigma_g^2$ for $k \in [M]$.*

**Assumption C.3** (Bounded Dissimilarity of $F(\mathbf{w})$)**.** *There exists constants $\beta^2 \ge 1$, $\kappa^2 \ge 0$ such that $\frac{1}{M}\sum_{i=1}^{M} \|\nabla F_i(\mathbf{w})\|^2 \le \beta^2 \|\frac{1}{M}\sum_{i=1}^{M}\nabla F_i(\mathbf{w})\|^2 + \kappa^2$ for any $\mathbf{w}$.*

Assumption C.1-C.3 are standard assumptions frequently used in the optimization literature [36, 3, 37, 4], including the $L_c$-continunity assumption [38, 39]. Note that we do not have any assumptions over our proposed objective function $\widetilde{F}(\mathbf{w})$ and only use the conventional assumptions used in FL for the standard objective function $F(\mathbf{w})$ to prove the convergence of INCFL over $\widetilde{F}(\mathbf{w})$ in Theorem C.1.

**Theorem C.1** (Convergence to the INCFL Objective $\widetilde{F}(\mathbf{w})$)**.** *Under Assumption C.1-C.3, suppose the server uniformly selects $m$ out of $M$ clients without replacement in each global round of Algorithm 1. With $\eta_l = \frac{1}{\sqrt{T}\tau}$, $\eta_g = \sqrt{\tau m}$, for a sufficiently large $T$ our optimization error is bounded as follows:*

$$\min_{t \in [T]} \mathbb{E}\left[\left\|\nabla \widetilde{F}(\mathbf{w}^{(t,0)})\right\|^2\right] \le \mathcal{O}\left(\frac{\sigma_g^2}{\sqrt{m\tau T}}\right) + \mathcal{O}\left(\frac{\sigma_g^2}{T\tau}\right) + \mathcal{O}\left(\frac{\sqrt{\tau}}{\sqrt{Tm}}\right) + \mathcal{O}\left(\frac{\kappa^2 + \beta^2}{T}\right) \quad (50)$$

*where $\mathcal{O}$ subsumes all constants (including $L_s$ and $L_c$).*

Theorem C.1 shows that with a sufficiently large number of communication rounds $T$ we reach a stationary point of our objective function $\widetilde{F}(\mathbf{w})$. The proof is deferred to Appendix D.2 where we also show a version of this theorem that contains the learning rates $\eta_g$ and $\eta_l$ with the constants.

# D Convergence Proof

## D.1 Preliminaries

First, we introduce the key lemmas used for the convergence analysis.

**Lemma D.1** (Bounded Dissimilarity for $\widetilde{F}(\mathbf{w})$)**.** *With Assumption C.1 and Assumption C.3 we have the bounded dissimilarity with respect to $\widetilde{F}(\mathbf{w})$ as:*

$$\frac{1}{M}\sum_{i=1}^{M}\|\nabla \widetilde{F}_i(\mathbf{w})\|^2 \le \beta'^2 \|\nabla \widetilde{F}(\mathbf{w})\|^2 + \kappa'^2 \quad (51)$$

*where $\beta'^2 = 2\beta^2$, $\kappa'^2 = 4\beta^2 L_c^2 + \kappa^2$*

*Proof.* One can easily show that

$$\frac{1}{M}\sum_{i=1}^{M}\|\nabla \widetilde{F}_i(\mathbf{w})\|^2 = \frac{1}{M}\sum_{i=1}^{M} q_i(\mathbf{w})^2 \|\nabla F_i(\mathbf{w})\|^2 \le \frac{1}{M}\sum_{i=1}^{M}\|\nabla F_i(\mathbf{w})\|^2 \quad (52)$$

due to $q_i(\mathbf{w}) \le 1$. Hence we have from Assumption C.3 and Cauchy-Schwarz inequality that

$$\frac{1}{M}\sum_{i=1}^{M}\|\nabla \widetilde{F}_i(\mathbf{w})\|^2 \le \frac{1}{M}\sum_{i=1}^{M}\|\nabla F_i(\mathbf{w})\|^2 \le \beta^2 \|\nabla F(\mathbf{w}) - \nabla \widetilde{F}(\mathbf{w}) + \nabla \widetilde{F}(\mathbf{w})\|^2 + \kappa^2 \quad (53)$$

$$\le 2\beta^2 \|\nabla F(\mathbf{w}) - \nabla \widetilde{F}(\mathbf{w})\|^2 + 2\beta^2 \|\nabla \widetilde{F}(\mathbf{w})\|^2 + \kappa^2 \quad (54)$$

We bound the first term in (54) as

$$\|\nabla F(\mathbf{w}) - \nabla \widetilde{F}(\mathbf{w})\|^2 = \left\|\sum_{i=1}^{M} \frac{(1 - q_i(\mathbf{w}))}{M} \nabla F_i(\mathbf{w})\right\|^2 \leq \frac{1}{M} \sum_{i=1}^{M} \|(1 - q_i(\mathbf{w})) \nabla F_i(\mathbf{w})\|^2 \quad (55)$$

$$\leq \frac{2}{M} \sum_{i=1}^{M} \|\nabla F_i(\mathbf{w})\|^2 \leq 2L_c^2 \quad (56)$$

where in (56) we use $q_i(\mathbf{w}) \leq 1, \forall i \in [M]$ and Assumption C.1. Then from (54) we have

$$\frac{1}{M} \sum_{i=1}^{M} \|\nabla \widetilde{F}_i(\mathbf{w})\|^2 \leq 2\beta^2 \|\nabla \widetilde{F}(\mathbf{w})\|^2 + \kappa^2 + 4\beta^2 L_c^2 \quad (57)$$

completing the proof. $\square$

**Lemma D.2** (Smoothness of $\widetilde{F}(\mathbf{w})$). *If Assumption C.1 is satisfied we have that the incentive local objectives, $\widetilde{F}_1(\mathbf{w}), \ldots, \widetilde{F}_M(\mathbf{w})$, are also $\widetilde{L}_s$-smooth for any $\mathbf{w}$ where $\widetilde{L}_s = L_c^2/4 + q_i(\mathbf{w})L_s$.*

*Proof.* Recall the definitions of $\widetilde{F}(\mathbf{w})$ below:

$$\widetilde{F}(\mathbf{w}) = \frac{1}{M} \sum_{i=1}^{M} \widetilde{F}_i(\mathbf{w}), \ \widetilde{F}_i(\mathbf{w}) := \sigma(F_i(\mathbf{w}) - F_i(\widehat{\mathbf{w}}_i^*)) \quad (58)$$

Let $\|\ \|_{op}$ denote the spectral norm of a matrix. Accordingly, with the model parameter vector $\mathbf{w} \in \mathbb{R}^d$, we have the spectral norm of the Hessian of $\widetilde{F}_i(\mathbf{w})$, $\forall i \in [M]$ as:

$$\|\nabla^2 \widetilde{F}_i(\mathbf{w})\|_{op} = \|q_i(\mathbf{w})[(\nabla F_i(\mathbf{w}) \nabla F_i(\mathbf{w})^T)(1 - q_i(\mathbf{w})) + \nabla^2 F_i(\mathbf{w})]\|_{op} \quad (59)$$

where $q_i(\mathbf{w}) = \text{Sigmoid}(F_i(\mathbf{w}) - F_i(\widehat{\mathbf{w}}_i^*))$ and $\nabla F_i(\mathbf{w}) \in \mathbb{R}^{d \times 1}$ is the gradient vector for the local objective $F_i(\mathbf{w})$ and $\nabla^2 F_i(\mathbf{w}) \in \mathbb{R}^{d \times d}$ is the Hessian of $F_i(\mathbf{w})$. We can bound the RHS of (59) as follows

$$\|\nabla^2 \widetilde{F}_i(\mathbf{w})\|_{op} = \|q_i(\mathbf{w})(1 - q_i(\mathbf{w}))(\nabla F_i(\mathbf{w}) \nabla F_i(\mathbf{w})^T) + q_i(\mathbf{w}) \nabla^2 F_i(\mathbf{w})\|_{op} \quad (60)$$

$$\leq \|q_i(\mathbf{w})(1 - q_i(\mathbf{w}))(\nabla F_i(\mathbf{w}) \nabla F_i(\mathbf{w})^T)\|_{op} + \|q_i(\mathbf{w}) \nabla^2 F_i(\mathbf{w})\|_{op} \quad (61)$$

$$= q_i(\mathbf{w})(1 - q_i(\mathbf{w}))\|(\nabla F_i(\mathbf{w}) \nabla F_i(\mathbf{w})^T)\|_{op} + q_i(\mathbf{w})\|\nabla^2 F_i(\mathbf{w})\|_{op} \quad (62)$$

$$= q_i(\mathbf{w})(1 - q_i(\mathbf{w}))\|\nabla F_i(\mathbf{w})\|^2 + q_i(\mathbf{w})\|\nabla^2 F_i(\mathbf{w})\|_{op} \quad (63)$$

$$\leq \frac{L_c^2}{4} + q_i(\mathbf{w})L_s \quad (64)$$

where we use triangle inequality in (61), and use $\|\mathbf{x}\mathbf{y}^T\|_{op} = \|\mathbf{x}\|\|\mathbf{y}\|$ in (63), and use $q_i(\mathbf{w}) \leq 1$ along with Assumption C.1 in (64). Since the norm of the Hessian of $\widetilde{F}_i(\mathbf{w})$ is bounded by $\frac{L_c^2}{4} + q_i(\mathbf{w})L_s$ we complete the proof. $\square$

### D.2  Proof of Theorem C.1 – Full Client Participation

For ease of writing, we define the following auxiliary variables for any client $i \in [M]$:

$$\text{Weighted Stochastic Gradient: } \mathbf{h}_i^{(t,0)} := q_i(\mathbf{w}^{(t,0)}) \sum_{r=0}^{\tau-1} \mathbf{g}(\mathbf{w}_i^{(t,r)}, \xi_i^{(t,r)}), \quad (65)$$

$$\text{Weighted Gradient: } \overline{\mathbf{h}}_i^{(t,0)} := q_i(\mathbf{w}^{(t,0)}) \sum_{r=0}^{\tau-1} \nabla F_i(\mathbf{w}_i^{(t,r)}), \quad (66)$$

$$\text{Normalized Global Learning Rate: } \eta_g^{(t,0)} := \eta_g / \left(\sum_{i=1}^{M} q_i(\mathbf{w}^{(t,0)}) + \epsilon\right) \quad (67)$$

where $\epsilon$ is a constant added to the denominator to prevent the denominator from being 0. From Algorithm 1 with full client participation, our proposed algorithm has the following effective update rule for the global model at the server:

$$\mathbf{w}^{(t+1,0)} = \mathbf{w}^{(t,0)} - \eta_g^{(t,0)} \eta_l \sum_{k=1}^{M} \mathbf{h}_k^{(t,0)} \tag{68}$$

With the update rule in (68), defining $\widetilde{\eta}^{(t,0)} := \eta_g^{(t,0)} \eta_l \tau M$ and using Lemma D.2 we have

$$\mathbb{E}\left[ \widetilde{F}(\mathbf{w}^{(t+1,0)}) \right] - \widetilde{F}(\mathbf{w}^{(t,0)}) \leq -\widetilde{\eta}^{(t,0)} \mathbb{E}\left[ \left\langle \nabla \widetilde{F}(\mathbf{w}^{(t,0)}), \frac{1}{M\tau} \sum_{i=1}^{M} \mathbf{h}_i^{(t,0)} \right\rangle \right]$$
$$+ \frac{\widetilde{L}_s(\widetilde{\eta}^{(t,0)})^2}{2} \mathbb{E}\left[ \left\| \frac{1}{M\tau} \sum_{i=1}^{M} \mathbf{h}_i^{(t,0)} \right\|^2 \right] \tag{69}$$

$$= -\widetilde{\eta}^{(t,0)} \mathbb{E}\left[ \left\langle \nabla \widetilde{F}(\mathbf{w}^{(t,0)}), \frac{1}{M\tau} \sum_{i=1}^{M} \left( \mathbf{h}_i^{(t,0)} - \overline{\mathbf{h}}_i^{(t,0)} \right) \right\rangle \right] - \widetilde{\eta}^{(t,0)} \mathbb{E}\left[ \left\langle \nabla \widetilde{F}(\mathbf{w}^{(t,0)}), \frac{1}{M\tau} \sum_{i=1}^{M} \overline{\mathbf{h}}_i^{(t,0)} \right\rangle \right]$$
$$+ \frac{\widetilde{L}_s(\widetilde{\eta}^{(t,0)})^2}{2} \mathbb{E}\left[ \left\| \frac{1}{M\tau} \sum_{i=1}^{M} \mathbf{h}_i^{(t,0)} \right\|^2 \right] \tag{70}$$

$$= -\frac{\widetilde{\eta}^{(t,0)}}{2} \left\| \nabla \widetilde{F}(\mathbf{w}^{(t,0)}) \right\|^2 - \frac{\widetilde{\eta}^{(t,0)}}{2} \mathbb{E}\left[ \left\| \frac{1}{M\tau} \sum_{i=1}^{M} \overline{\mathbf{h}}_i^{(t,0)} \right\|^2 \right] + \frac{\widetilde{\eta}^{(t,0)}}{2} \mathbb{E}\left[ \left\| \nabla \widetilde{F}(\mathbf{w}^{(t,0)}) - \frac{1}{M\tau} \sum_{i=1}^{M} \overline{\mathbf{h}}_i^{(t,0)} \right\|^2 \right]$$
$$+ \frac{\widetilde{L}_s(\widetilde{\eta}^{(t,0)})^2}{2M^2\tau^2} \mathbb{E}\left[ \left\| \sum_{i=1}^{M} \mathbf{h}_i^{(t,0)} \right\|^2 \right] \tag{71}$$

For the last term in (71), we can bound it as

$$\frac{\widetilde{L}_s(\widetilde{\eta}^{(t,0)})^2}{2M^2\tau^2} \mathbb{E}\left[ \left\| \sum_{i=1}^{M} \mathbf{h}_i^{(t,0)} \right\|^2 \right] \leq \frac{\widetilde{L}_s(\widetilde{\eta}^{(t,0)})^2}{M^2\tau^2} \sum_{i=1}^{M} \mathbb{E}\left[ \left\| \mathbf{h}_i^{(t,0)} - \overline{\mathbf{h}}_i^{(t,0)} \right\|^2 \right] + \frac{\widetilde{L}_s(\widetilde{\eta}^{(t,0)})^2}{M^2\tau^2} \mathbb{E}\left[ \left\| \sum_{i=1}^{M} \overline{\mathbf{h}}_i^{(t,0)} \right\|^2 \right] \tag{72}$$

$$= \frac{\widetilde{L}_s(\widetilde{\eta}^{(t,0)})^2}{M^2\tau^2} \sum_{i=1}^{M} \mathbb{E}\left[ \left\| q_i(\mathbf{w}^{(t,0)}) \sum_{r=0}^{\tau-1} \left( \mathbf{g}(\mathbf{w}_i^{(t,r)}, \xi_i^{(t,r)}) - \nabla F_i(\mathbf{w}_i^{(t,r)}) \right) \right\|^2 \right] + \frac{\widetilde{L}_s(\widetilde{\eta}^{(t,0)})^2}{M^2\tau^2} \mathbb{E}\left[ \left\| \sum_{i=1}^{M} \overline{\mathbf{h}}_i^{(t,0)} \right\|^2 \right] \tag{73}$$

$$= \frac{\widetilde{L}_s(\widetilde{\eta}^{(t,0)})^2}{M^2\tau^2} \sum_{i=1}^{M} q_i(\mathbf{w}^{(t,0)})^2 \sum_{r=0}^{\tau-1} \mathbb{E}\left[ \left\| \mathbf{g}(\mathbf{w}_i^{(t,r)}, \xi_i^{(t,r)}) - \nabla F_i(\mathbf{w}_i^{(t,r)}) \right\|^2 \right] + \frac{\widetilde{L}_s(\widetilde{\eta}^{(t,0)})^2}{M^2\tau^2} \mathbb{E}\left[ \left\| \sum_{i=1}^{M} \overline{\mathbf{h}}_i^{(t,0)} \right\|^2 \right] \tag{74}$$

$$= \frac{\widetilde{L}_s(\widetilde{\eta}^{(t,0)})^2}{M^2\tau^2} \sum_{i=1}^{M} q_i(\mathbf{w}^{(t,0)})^2 \tau \sigma_g^2 + \frac{\widetilde{L}_s(\widetilde{\eta}^{(t,0)})^2}{M^2\tau^2} \mathbb{E}\left[ \left\| \sum_{i=1}^{M} \overline{\mathbf{h}}_i^{(t,0)} \right\|^2 \right] \tag{75}$$

$$\leq \frac{\widetilde{L}_s(\widetilde{\eta}^{(t,0)})^2 \sigma_g^2}{M\tau} + \widetilde{L}_s(\widetilde{\eta}^{(t,0)})^2 \mathbb{E}\left[ \left\| \frac{1}{M\tau} \sum_{i=1}^{M} \overline{\mathbf{h}}_i^{(t,0)} \right\|^2 \right] \tag{76}$$

where (72) is due to the Cauchy-Schwartz inequality and (75) is due to Assumption C.2 and (76) is
due to $q_i(\mathbf{w}) \leq 1, \forall i \in [M]$. Merging (76) into (71) we have

$$
\mathbb{E}\left[\widetilde{F}(\mathbf{w}^{(t+1,0)})\right] - \widetilde{F}(\mathbf{w}^{(t,0)}) \leq -\frac{\widetilde{\eta}^{(t,0)}}{2}\left\|\nabla\widetilde{F}(\mathbf{w}^{(t,0)})\right\|^2 + \frac{\widetilde{\eta}^{(t,0)}}{2}\mathbb{E}\left[\left\|\nabla\widetilde{F}(\mathbf{w}^{(t,0)}) - \frac{1}{M\tau}\sum_{i=1}^{M}\overline{\mathbf{h}}_i^{(t,0)}\right\|^2\right]
$$
$$
+ \frac{\widetilde{L}_s(\widetilde{\eta}^{(t,0)})^2\sigma_g^2}{M\tau} + \left((\widetilde{\eta}^{(t,0)})^2\widetilde{L}_s - \frac{\widetilde{\eta}^{(t,0)}}{2}\right)\mathbb{E}\left[\left\|\frac{1}{M\tau}\sum_{i=1}^{M}\overline{\mathbf{h}}_i^{(t,0)}\right\|^2\right]
$$
$$
\tag{77}
$$

Now we aim at bounding the second term in the RHS of (77) as follows:

$$
\frac{\widetilde{\eta}^{(t,0)}}{2}\mathbb{E}\left[\left\|\nabla\widetilde{F}(\mathbf{w}^{(t,0)}) - \frac{1}{M\tau}\sum_{i=1}^{M}\overline{\mathbf{h}}_i^{(t,0)}\right\|^2\right] \tag{78}
$$

$$
= \frac{\widetilde{\eta}^{(t,0)}}{2}\mathbb{E}\left[\left\|\frac{1}{M}\sum_{i=1}^{M}q_i(\mathbf{w}^{(t,0)})\nabla F_i(\mathbf{w}^{(t,0)}) - \frac{1}{M\tau}\sum_{i=1}^{M}q_i(\mathbf{w}^{(t,0)})\sum_{r=0}^{\tau-1}\nabla F_i(\mathbf{w}_i^{(t,r)})\right\|^2\right] \tag{79}
$$

$$
= \frac{\widetilde{\eta}^{(t,0)}}{2}\mathbb{E}\left[\left\|\frac{1}{M\tau}\sum_{i=1}^{M}q_i(\mathbf{w}^{(t,0)})\sum_{r=0}^{\tau-1}\left(\nabla F_i(\mathbf{w}^{(t,0)}) - \nabla F_i(\mathbf{w}_i^{(t,r)})\right)\right\|^2\right] \tag{80}
$$

$$
\leq \frac{\widetilde{\eta}^{(t,0)}}{2M\tau}\sum_{i=1}^{M}q_i(\mathbf{w}^{(t,0)})^2\sum_{r=0}^{\tau-1}\mathbb{E}\left[\left\|\nabla F_i(\mathbf{w}^{(t,0)}) - \nabla F_i(\mathbf{w}_i^{(t,r)})\right\|^2\right] \tag{81}
$$

$$
= \frac{L_s^2\widetilde{\eta}^{(t,0)}}{2M\tau}\sum_{i=1}^{M}q_i(\mathbf{w}^{(t,0)})^2\sum_{r=0}^{\tau-1}\mathbb{E}\left[\left\|\mathbf{w}^{(t,0)} - \mathbf{w}_i^{(t,r)}\right\|^2\right] \tag{82}
$$

where (81) is due to Jensen's inequality and (82) is due to Lemma D.2. We can bound the difference
of the global model and local model for any client $i \in [M]$ as follows:

$$
\mathbb{E}\left[\left\|\mathbf{w}^{(t,0)} - \mathbf{w}_i^{(t,r)}\right\|^2\right] = \eta_l^2\mathbb{E}\left[\left\|\sum_{l=0}^{r-1}\mathbf{g}(\mathbf{w}_i^{(t,l)}, \xi_i^{(t,l)})\right\|^2\right] \tag{83}
$$

$$
\leq 2\eta_l^2\mathbb{E}\left[\left\|\sum_{l=0}^{r-1}\mathbf{g}(\mathbf{w}_i^{(t,l)}, \xi_i^{(t,l)}) - \nabla F_i(\mathbf{w}_i^{(t,l)})\right\|^2\right] + 2\eta_l^2\mathbb{E}\left[\left\|\sum_{l=0}^{r-1}\nabla F_i(\mathbf{w}_i^{(t,l)})\right\|^2\right] \tag{84}
$$

$$
\leq 2\eta_l^2\sigma_g^2 r + 2\eta_l^2\mathbb{E}\left[\left\|\sum_{l=0}^{r-1}\nabla F_i(\mathbf{w}_i^{(t,l)})\right\|^2\right] \tag{85}
$$

where (84) is due to Cauchy-Schwarz inequality and (85) is due to Assumption C.2. We bound the
last term in (85) as follows:

$$
\mathbb{E}\left[\left\|\sum_{l=0}^{r-1}\nabla F_i(\mathbf{w}_i^{(t,l)})\right\|^2\right] \leq r\sum_{l=0}^{r-1}\mathbb{E}\left[\left\|\nabla F_i(\mathbf{w}_i^{(t,l)})\right\|^2\right] \leq \tau\sum_{l=0}^{\tau-1}\mathbb{E}\left[\left\|\nabla F_i(\mathbf{w}_i^{(t,l)})\right\|^2\right] \tag{86}
$$

$$
\leq 2\tau\sum_{l=0}^{\tau-1}\mathbb{E}\left[\left\|\nabla F_i(\mathbf{w}_i^{(t,l)}) - \nabla F_i(\mathbf{w}^{(t,0)})\right\|^2\right] + 2\tau^2\mathbb{E}\left[\left\|\nabla F_i(\mathbf{w}^{(t,0)})\right\|^2\right] \tag{87}
$$

$$
\leq 2L_s^2\tau\sum_{l=0}^{\tau-1}\mathbb{E}\left[\left\|\mathbf{w}_i^{(t,l)} - \mathbf{w}^{(t,0)}\right\|^2\right] + 2\tau^2\mathbb{E}\left[\left\|\nabla F_i(\mathbf{w}^{(t,0)})\right\|^2\right] \tag{88}
$$

where (86) is due to Jensen's inequality, and (87) is due to Cauchy-Schwarz inequality, and (88) is due to Lemma D.2. Combining (88) with (85) we have that

$$\mathbb{E}\left[\left\|\mathbf{w}^{(t,0)} - \mathbf{w}_i^{(t,r)}\right\|^2\right] \leq 2\eta_l^2\sigma_g^2 r + 4L_s^2\eta_l^2\tau\sum_{l=0}^{\tau-1}\mathbb{E}\left[\left\|\mathbf{w}^{(t,0)} - \mathbf{w}_i^{(t,l)}\right\|^2\right] + 4\eta_l^2\tau^2\mathbb{E}\left[\left\|\nabla F_i(\mathbf{w}^{(t,0)})\right\|^2\right]$$

(89)

Reorganizing (89) and taking the summation $r \in [\tau]$ on both sides we have,

$$(1 - 4L_s^2\eta_l^2\tau^2)\sum_{r=0}^{\tau-1}\mathbb{E}\left[\left\|\mathbf{w}^{(t,0)} - \mathbf{w}_i^{(t,r)}\right\|^2\right] \leq 2\eta_l^2\sigma_g^2\sum_{r=0}^{\tau-1}r + 4\eta_l^2\tau^3\mathbb{E}\left[\left\|\nabla F_i(\mathbf{w}^{(t,0)})\right\|^2\right] \quad (90)$$

$$\leq \eta_l^2\sigma_g^2\tau^2 + 4\eta_l^2\tau^3\mathbb{E}\left[\left\|\nabla F_i(\mathbf{w}^{(t,0)})\right\|^2\right] \quad (91)$$

With $\eta_l \leq 1/(2\sqrt{2}\tau L_s)$, we have that $1/(1 - 4L_s^2\eta_l^2\tau^2) \leq 2$ and hence can further bound (91) as

$$\sum_{r=0}^{\tau-1}\mathbb{E}\left[\left\|\mathbf{w}^{(t,0)} - \mathbf{w}_i^{(t,r)}\right\|^2\right] \leq 2\eta_l^2\sigma_g^2\tau^2 + 8\eta_l^2\tau^3\mathbb{E}\left[\left\|\nabla F_i(\mathbf{w}^{(t,0)})\right\|^2\right] \quad (92)$$

Finally, plugging in (92) to (82) we have

$$\frac{\widetilde{\eta}^{(t,0)}}{2}\mathbb{E}\left[\left\|\nabla\widetilde{F}(\mathbf{w}^{(t,0)}) - \frac{1}{M\tau}\sum_{i=1}^{M}\overline{\mathbf{h}}_i^{(t,0)}\right\|^2\right]$$

(93)

$$\leq \frac{L_s^2\widetilde{\eta}^{(t,0)}}{2M\tau}\sum_{i=1}^{M}q_i(\mathbf{w}^{(t,0)})^2\left(2\eta_l^2\sigma_g^2\tau^2 + 8\eta_l^2\tau^3\mathbb{E}\left[\left\|\nabla F_i(\mathbf{w}^{(t,0)})\right\|^2\right]\right)$$

$$\leq L_s^2\widetilde{\eta}^{(t,0)}\eta_l^2\sigma_g^2\tau + 4\eta_l^2\tau^2 L_s^2\widetilde{\eta}^{(t,0)}\frac{1}{M}\sum_{i=1}^{M}\mathbb{E}\left[\left\|\nabla F_i(\mathbf{w}^{(t,0)})\right\|^2\right] \quad (94)$$

$$\leq L_s^2\widetilde{\eta}^{(t,0)}\eta_l^2\sigma_g^2\tau + 4\eta_l^2\tau^2 L_s^2\widetilde{\eta}^{(t,0)}(\beta'^2\left\|\nabla\widetilde{F}(\mathbf{w}^{(t,0)})\right\|^2 + \kappa'^2) \quad (95)$$

where (94) uses $q_i(\mathbf{w}) \leq 1, \forall i \in [M]$ and (95) uses Lemma D.1. Merging (95) to (77) we have

$$\mathbb{E}\left[\widetilde{F}(\mathbf{w}^{(t+1,0)})\right] - \widetilde{F}(\mathbf{w}^{(t,0)})$$

$$\leq -\frac{\widetilde{\eta}^{(t,0)}}{2}\left\|\nabla\widetilde{F}(\mathbf{w}^{(t,0)})\right\|^2 + \widetilde{\eta}^{(t,0)}\left(\widetilde{\eta}^{(t,0)}\widetilde{L}_s - \frac{1}{2}\right)\mathbb{E}\left[\left\|\frac{1}{M\tau}\sum_{i=1}^{M}\overline{\mathbf{h}}_i^{(t,0)}\right\|^2\right] \quad (96)$$

$$+\frac{\widetilde{L}_s(\widetilde{\eta}^{(t,0)})^2\sigma_g^2}{M\tau} + \widetilde{\eta}^{(t,0)}L_s^2\eta_l^2\sigma_g^2\tau + 4\widetilde{\eta}^{(t,0)}\eta_l^2\tau^2 L_s^2\beta'^2\left\|\nabla\widetilde{F}(\mathbf{w}^{(t,0)})\right\|^2 + 4\widetilde{\eta}^{(t,0)}\eta_l^2\tau^2 L_s^2\kappa'^2$$

With $\eta_l\eta_g \leq 1/(4\tau L_s)$ we have that $\widetilde{\eta}^{(t,0)}\widetilde{L}_s - \frac{1}{2} \leq -1/4$ and thus can further simplify (96) to

$$\mathbb{E}\left[\widetilde{F}(\mathbf{w}^{(t+1,0)})\right] - \widetilde{F}(\mathbf{w}^{(t,0)}) \leq -\frac{\widetilde{\eta}^{(t,0)}}{2}\left\|\nabla\widetilde{F}(\mathbf{w}^{(t,0)})\right\|^2 + 4\widetilde{\eta}^{(t,0)}\eta_l^2\tau^2 L_s^2\beta'^2\left\|\nabla\widetilde{F}(\mathbf{w}^{(t,0)})\right\|^2$$

$$+\frac{\widetilde{L}_s(\widetilde{\eta}^{(t,0)})^2\sigma_g^2}{M\tau} + \widetilde{\eta}^{(t,0)}L_s^2\eta_l^2\sigma_g^2\tau + 4\widetilde{\eta}^{(t,0)}\eta_l^2\tau^2 L_s^2\kappa'^2$$

(97)

$$= \widetilde{\eta}^{(t,0)}\left(4\eta_l^2\tau^2 L_s^2\beta' - \frac{1}{2}\right)\left\|\nabla\widetilde{F}(\mathbf{w}^{(t,0)})\right\|^2 + \frac{\widetilde{L}_s(\widetilde{\eta}^{(t,0)})^2\sigma_g^2}{M\tau} + \widetilde{\eta}^{(t,0)}L_s^2\eta_l^2\sigma_g^2\tau + 4\widetilde{\eta}^{(t,0)}\eta_l^2\tau^2 L_s^2\kappa'^2$$

(98)

With local learning rate $\eta_l \leq \min\{1/(4\tau L_s), 1/(4\beta'\tau L_s)\}$ we have that

$$\mathbb{E}\left[\widetilde{F}(\mathbf{w}^{(t+1,0)})\right] - \widetilde{F}(\mathbf{w}^{(t,0)}) \leq -\frac{\widetilde{\eta}^{(t,0)}}{4}\left\|\nabla\widetilde{F}(\mathbf{w}^{(t,0)})\right\|^2 + \frac{\widetilde{L}_s(\widetilde{\eta}^{(t,0)})^2\sigma_g^2}{M\tau} + \widetilde{\eta}^{(t,0)}L_s^2\eta_l^2\sigma_g^2\tau$$

$$+4\widetilde{\eta}^{(t,0)}\eta_l^2\tau^2 L_s^2\kappa'^2$$

(99)

and we use the property of $\widetilde{\eta}^{(t,0)}$ that $\frac{M\tau\eta_l\eta_g}{M+\epsilon} \leq \widetilde{\eta}^{(t,0)} \leq \frac{M\tau\eta_l\eta_g}{\epsilon}$ to get

$$\mathbb{E}\left[\widetilde{F}(\mathbf{w}^{(t+1,0)})\right] - \widetilde{F}(\mathbf{w}^{(t,0)}) \leq -\frac{M\tau\eta_l\eta_g}{4(M+\epsilon)}\left\|\nabla\widetilde{F}(\mathbf{w}^{(t,0)})\right\|^2 + \frac{\widetilde{L}_s M\tau\eta_l^2\eta_g^2\sigma_g^2}{\epsilon^2}$$
$$+ \frac{M\tau^2 L_s^2\eta_l^3\eta_g\sigma_g^2}{\epsilon} + \frac{4M\eta_l^3\eta_g\tau^3 L_s^2\kappa'^2}{\epsilon} \tag{100}$$

Taking the average across all rounds on both sides of (100) we get

$$\frac{1}{T}\sum_{t=0}^{T-1}\mathbb{E}\left[\|\nabla\widetilde{F}(\mathbf{w}^{(t,0)})\|^2\right] \leq \frac{4(M+\epsilon)\left(\widetilde{F}(\mathbf{w}^{(0,0)}) - \widetilde{F}_{\text{inf}}\right)}{M\tau\eta_l\eta_g T} + \frac{16\eta_l^2\tau^2 L_s^2\kappa'^2(M+\epsilon)}{\epsilon}$$
$$+ \frac{4L_s^2\eta_l^2\tau\sigma_g^2(M+\epsilon)}{\epsilon} + \frac{4\eta_g\eta_l\widetilde{L}_s\sigma_g^2(M+\epsilon)}{\epsilon^2} \tag{101}$$

and prove

$$\min_{t\in[T]}\mathbb{E}\left[\left\|\nabla\widetilde{F}(\mathbf{w}^{(t,0)})\right\|^2\right] \leq \frac{1}{T}\sum_{t=0}^{T-1}\mathbb{E}\left[\|\nabla\widetilde{F}(\mathbf{w}^{(t,0)})\|^2\right] \leq \frac{4(M+\epsilon)\left(\widetilde{F}(\mathbf{w}^{(0,0)}) - \widetilde{F}_{\text{inf}}\right)}{M\tau\eta_l\eta_g T}$$
$$+ \frac{16\eta_l^2\tau^2 L_s^2\kappa'^2(M+\epsilon)}{\epsilon} + \frac{4L_s^2\eta_l^2\tau\sigma_g^2(M+\epsilon)}{\epsilon} + \frac{4\eta_g\eta_l\widetilde{L}_s\sigma_g^2(M+\epsilon)}{\epsilon^2} \tag{102}$$

Further, using $\tilde{L}_s = \frac{L_s}{M}\sum_{k=1}^M q_k(\mathbf{w}) + \frac{L_c}{4}$ and $\epsilon = \frac{ML_c}{4L_s} > 0$ from the optimal learning rate we have the bound in (102) to be

$$\min_{t\in[T]}\mathbb{E}\left[\left\|\nabla\widetilde{F}(\mathbf{w}^{(t,0)})\right\|^2\right] \leq \frac{(4L_s+L_c)\left(\widetilde{F}(\mathbf{w}^{(0,0)}) - \widetilde{F}_{\text{inf}}\right)}{L_s\tau\eta_l\eta_g T} + \frac{64\eta_l^2\tau^2 L_s^2\kappa'^2(4L_s+L_c)}{L_c}$$
$$+ \frac{4L_s^2\eta_l^2\tau\sigma_g^2(4L_s+L_c)}{L_c} + \frac{64L_s\eta_g\eta_l\sigma_g^2(L_s+L_c/4)^2}{ML_c^2} \tag{103}$$

By setting the global and local learning rate as $\eta_g = \sqrt{\tau M}$ and $\eta_l = \frac{1}{\sqrt{T}\tau}$ we can further optimize the bound as

$$\min_{t\in[T]}\mathbb{E}\left[\left\|\nabla\widetilde{F}(\mathbf{w}^{(t,0)})\right\|^2\right] \leq \frac{(4L_s+L_c)\left(\widetilde{F}(\mathbf{w}^{(0,0)}) - \widetilde{F}_{\text{inf}}\right)}{L_s\sqrt{TM\tau}} + \frac{64L_s^2\kappa'^2(4L_s+L_c)}{L_c T}$$
$$+ \frac{4L_s^2\sigma_g^2(4L_s+L_c)}{T\tau L_c} + \frac{64L_s\sigma_g^2(L_s+L_c/4)^2}{\sqrt{TM\tau}} \tag{104}$$

completing the full client participation proof of Theorem C.1.

### D.3 Proof of Theorem C.1 – Partial Client Participation

We present the convergence guarantees of INCFL for partical client participation in this section. With partical client participation, we have the update rule in (68) changed to

$$\mathbf{w}^{(t+1,0)} = \mathbf{w}^{(t,0)} - \eta_g^{(t,0)}\eta_l\sum_{k\in\mathcal{S}^{(t,0)}}\mathbf{h}_k^{(t,0)} \tag{105}$$

where the $m$ clients are sampled uniformly at random without replacement for $\mathcal{S}^{(t,0)}$ at each communication round $t$ by the server and $\eta_g^{(t,0)} = m\eta_g/(\sum_{k\in\mathcal{S}^{(t,0)}} q_k(\mathbf{w}^{(t,0)}) + \epsilon)$ for positive constant $\epsilon$. Then with the update rule in (105) and Lemma D.2, defining $\widetilde{\eta}^{(t,0)} = \eta_g^{(t,0)}\eta_l\tau m$ we have

$$\mathbb{E}\left[\widetilde{F}(\mathbf{w}^{(t+1,0)})\right] - \widetilde{F}(\mathbf{w}^{(t,0)}) \leq \mathbb{E}\left[-\widetilde{\eta}^{(t,0)}\left\langle\nabla\widetilde{F}(\mathbf{w}^{(t,0)}), \frac{1}{m\tau}\sum_{i\in\mathcal{S}^{(t,0)}}\mathbf{h}_i^{(t,0)}\right\rangle\right]$$
$$+ \mathbb{E}\left[\frac{\widetilde{L}_s(\widetilde{\eta}^{(t,0)})^2}{2}\left\|\frac{1}{m\tau}\sum_{i\in\mathcal{S}^{(t,0)}}\mathbf{h}_i^{(t,0)}\right\|^2\right] \tag{106}$$

For the first term in the RHS of (106) we have that due to the uniform sampling of clients (see Lemma 4 in [40]), it becomes analogous to the derivation for full client participation. Hence, with the property of $\frac{m\tau\eta_l\eta_g}{m+\epsilon} \leq \widetilde{\eta}^{(t,0)} \leq \frac{m\tau\eta_l\eta_g}{\epsilon}$ and using the previous bounds in (95), we result in the final bound for the first term in the RHS of (106) as below:

$$\mathbb{E}\left[-\widetilde{\eta}^{(t,0)}\left\langle \nabla\widetilde{F}(\mathbf{w}^{(t,0)}), \frac{1}{m\tau}\sum_{i\in\mathcal{S}^{(t,0)}}\mathbf{h}_i^{(t,0)}\right\rangle\right] \leq \left(-\frac{m\tau\eta_l\eta_g}{m+\epsilon} + \frac{4\eta_l^3\tau^3 L_s^2\beta'^2\eta_g m}{\epsilon}\right)\left\|\nabla\widetilde{F}(\mathbf{w}^{(t,0)})\right\|^2$$

$$+\frac{4L_s^2\tau^3\eta_l^3 m\eta_g\kappa'^2}{\epsilon} + \frac{L_s^2\tau^2\eta_l^2 m\eta_g\sigma_g^2}{\epsilon} \tag{107}$$

For the second term in the RHS of (106), with $C = \widetilde{L}_s(m\tau\eta_l\eta_g/\epsilon)^2$ we have the following:

$$\mathbb{E}\left[\frac{\widetilde{L}_s(\widetilde{\eta}^{(t,0)})^2}{2}\left\|\frac{1}{m\tau}\sum_{i\in\mathcal{S}^{(t,0)}}\mathbf{h}_i^{(t,0)}\right\|^2\right] \leq C\mathbb{E}\left[\left\|\frac{1}{m\tau}\sum_{i\in\mathcal{S}^{(t,0)}}(\mathbf{h}_i^{(t,0)} - \overline{\mathbf{h}}_i^{(t,0)})\right\|^2\right]$$

$$+C\mathbb{E}\left[\left\|\frac{1}{m\tau}\sum_{i\in\mathcal{S}^{(t,0)}}\overline{\mathbf{h}}_i^{(t,0)}\right\|^2\right] \tag{108}$$

$$= \frac{C}{m^2\tau^2}\mathbb{E}\left[\sum_{i\in\mathcal{S}^{(t,0)}}\left\|\mathbf{h}_i^{(t,0)} - \overline{\mathbf{h}}_i^{(t,0)}\right\|^2\right] + C\mathbb{E}\left[\left\|\frac{1}{m\tau}\sum_{i\in\mathcal{S}^{(t,0)}}\overline{\mathbf{h}}_i^{(t,0)}\right\|^2\right] \tag{109}$$

$$= \frac{C}{mM\tau^2}\sum_{i=1}^M\mathbb{E}\left[\left\|\mathbf{h}_i^{(t,0)} - \overline{\mathbf{h}}_i^{(t,0)}\right\|^2\right] + C\mathbb{E}\left[\left\|\frac{1}{m\tau}\sum_{i\in\mathcal{S}^{(t,0)}}\overline{\mathbf{h}}_i^{(t,0)}\right\|^2\right] \tag{110}$$

$$\leq \frac{C\sigma_g^2}{m\tau} + C\mathbb{E}\left[\left\|\frac{1}{m\tau}\sum_{i\in\mathcal{S}^{(t,0)}}\overline{\mathbf{h}}_i^{(t,0)}\right\|^2\right] \tag{111}$$

where (110) follows due to, again, the uniform sampling of clients and the rest follows identical steps for full client participation in the derivation for (72). Note that

$$C = \left(\frac{L_s}{M}\sum_{k=1}^M q_k(\mathbf{w}) + \frac{L_c}{4}\right)(m\tau\eta_l\eta_g/\epsilon)^2 \leq (L_s + \frac{L_c}{4})(m\tau\eta_l\eta_g/\epsilon)^2 \tag{112}$$

For the second term in (111) we have that

$$\mathbb{E}\left[\left\|\frac{1}{m\tau}\sum_{i\in\mathcal{S}^{(t,0)}}\overline{\mathbf{h}}_i^{(t,0)}\right\|^2\right] = \mathbb{E}\left[\left\|\frac{1}{m\tau}\sum_{i\in\mathcal{S}^{(t,0)}}\left(\overline{\mathbf{h}}_i^{(t,0)} - \nabla\widetilde{F}_i(\mathbf{w}^{(t,0)}) + \nabla\widetilde{F}_i(\mathbf{w}^{(t,0)})\right)\right.\right.$$

$$\left.\left.-\frac{1}{\tau}\nabla\widetilde{F}(\mathbf{w}^{(t,0)}) + \frac{1}{\tau}\nabla\widetilde{F}(\mathbf{w}^{(t,0)})\right\|^2\right] \tag{113}$$

$$\leq \underbrace{3\mathbb{E}\left[\left\|\frac{1}{m\tau}\sum_{i\in\mathcal{S}^{(t,0)}}\left(\overline{\mathbf{h}}_i^{(t,0)} - \nabla\widetilde{F}_i(\mathbf{w}^{(t,0)})\right)\right\|^2\right]}_{A_1} + \underbrace{\frac{3}{\tau^2}\mathbb{E}\left[\left\|\frac{1}{m}\sum_{i\in\mathcal{S}^{(t,0)}}\nabla\widetilde{F}_i(\mathbf{w}^{(t,0)}) - \nabla\widetilde{F}(\mathbf{w}^{(t,0)})\right\|^2\right]}_{A_2}$$

$$+3\mathbb{E}\left[\left\|\frac{1}{\tau}\nabla\widetilde{F}(\mathbf{w}^{(t,0)})\right\|^2\right] \tag{114}$$

First we bound $A_1$ in (114) as follows:

$$3\mathbb{E}\left[\left\|\frac{1}{m\tau}\sum_{i\in\mathcal{S}^{(t,0)}}\left(\overline{\mathbf{h}}_i^{(t,0)}-\nabla\widetilde{F}_i(\mathbf{w}^{(t,0)})\right)\right\|^2\right] \tag{115}$$

$$= 3\mathbb{E}\left[\left\|\frac{1}{m\tau}\sum_{i\in\mathcal{S}^{(t,0)}}q_i(\mathbf{w}^{(t,0)})\sum_{r=0}^{\tau-1}\left(\nabla F_i(\mathbf{w}_i^{(t,r)})-\nabla F_i(\mathbf{w}^{(t,0)})\right)\right\|^2\right]$$

$$\leq \frac{3}{m\tau}\mathbb{E}\left[\sum_{i\in\mathcal{S}^{(t,0)}}\sum_{r=0}^{\tau-1}\left\|\nabla F_i(\mathbf{w}_i^{(t,r)})-\nabla F_i(\mathbf{w}^{(t,0)})\right\|^2\right] \tag{116}$$

$$= \frac{3}{M\tau}\sum_{i=1}^{M}\sum_{r=0}^{\tau-1}\mathbb{E}\left[\left\|\nabla F_i(\mathbf{w}_i^{(t,r)})-\nabla F_i(\mathbf{w}^{(t,0)})\right\|^2\right] \tag{117}$$

$$\leq \frac{3L_s^2}{M\tau}\sum_{i=1}^{M}\sum_{r=0}^{\tau-1}\mathbb{E}\left[\left\|\mathbf{w}^{(t,0)}-\mathbf{w}_i^{(t,r)}\right\|^2\right] \tag{118}$$

where (116) is due to Jensen's inequality and $q_i(\mathbf{w}) \leq 1$ and (117) is due to the uniform sampling of clients, and (118) is due to Assumption C.1. Using (77) we have already derived, bound (118) further to:

$$3\mathbb{E}\left[\left\|\frac{1}{m\tau}\sum_{i\in\mathcal{S}^{(t,0)}}\left(\overline{\mathbf{h}}_i^{(t,0)}-\nabla\widetilde{F}_i(\mathbf{w}^{(t,0)})\right)\right\|^2\right] \leq 6L_s^2\eta_l^2\sigma_g^2\tau + \frac{24L_s^2\eta_l^2\tau^2}{M}\sum_{i=1}^{M}\mathbb{E}\left[\left\|\nabla F_i(\mathbf{w}^{(t,0)})\right\|^2\right] \tag{119}$$

$$\leq 6L_s^2\eta_l^2\sigma_g^2\tau + 24L_s^2\eta_l^2\tau^2(\beta'^2\|\nabla\widetilde{F}(\mathbf{w}^{(t,0)})\|^2+\kappa'^2) \tag{120}$$

where (120) is due to Lemma D.1.

Next we bound $A_2$ as follows:

$$\frac{3}{\tau^2}\mathbb{E}\left[\left\|\frac{1}{m}\sum_{i\in\mathcal{S}^{(t,0)}}\nabla\widetilde{F}_i(\mathbf{w}^{(t,0)})-\nabla\widetilde{F}(\mathbf{w}^{(t,0)})\right\|^2\right] \tag{121}$$

$$= \frac{3(M-m)}{\tau^2 mM(M-1)}\sum_{i=1}^{M}\mathbb{E}\left[\left\|\nabla\widetilde{F}_i(\mathbf{w}^{(t,0)})-\nabla\widetilde{F}(\mathbf{w}^{(t,0)})\right\|^2\right]$$

$$= \frac{3(M-m)}{\tau^2 mM(M-1)}\sum_{i=1}^{M}\left\|\nabla q_i(\mathbf{w}^{(t,0)})F_i(\mathbf{w}^{(t,0)})-\frac{1}{M}\sum_{i=1}^{M}q_i(\mathbf{w}^{(t,0)})\nabla F_i(\mathbf{w}^{(t,0)})\right\|^2 \tag{122}$$

$$\leq \frac{6(M-m)}{\tau^2 mM(M-1)}\sum_{i=1}^{M}\left(\left\|\nabla F_i(\mathbf{w}^{(t,0)})\right\|^2+\left\|\frac{1}{M}\sum_{i=1}^{M}q_i(\mathbf{w}^{(t,0)})\nabla F_i(\mathbf{w}^{(t,0)})\right\|^2\right) \tag{123}$$

$$\leq \frac{12(M-m)L_c^2}{\tau^2 m(M-1)} \tag{124}$$

where (121) is due to the variance under uniform sampling without replacement (see Lemma 4 in [40]) and (123) is due to the Cauchy-Schwarz inequality and (124) is due to Assumption C.1.

655 Mering the bounds for $A_1$ and $A_2$ to (114) we have that

$$
\mathbb{E}\left[\left\|\frac{1}{m\tau}\sum_{i\in\mathcal{S}^{(t,0)}}\overline{\mathbf{h}}_i^{(t,0)}\right\|^2\right] \leq 6L_s^2\eta_l^2\sigma_g^2\tau + 24L_s^2\eta_l^2\tau^2\beta'^2\|\nabla\widetilde{F}(\mathbf{w}^{(t,0)})\|^2 \tag{125}
$$

$$
+24L_s^2\eta_l^2\tau^2\kappa'^2 + \frac{12(M-m)L_c^2}{\tau^2 m(M-1)} + 3\mathbb{E}\left[\left\|\frac{1}{\tau}\nabla\widetilde{F}(\mathbf{w}^{(t,0)})\right\|^2\right]
$$

$$
= \left(24L_s^2\eta_l^2\tau^2\beta'^2 + \frac{3}{\tau^2}\right)\|\nabla\widetilde{F}(\mathbf{w}^{(t,0)})\|^2 + 6L_s^2\eta_l^2\tau(\sigma_g^2 + 4\tau\kappa'^2) + \frac{12(M-m)L_c^2}{\tau^2 m(M-1)} \tag{126}
$$

656 Then we can plug in (126) back to (111) and plugging in (107) to (106), we can derive the bound in
657 (106) as

$$
\mathbb{E}\left[\widetilde{F}(\mathbf{w}^{(t+1,0)})\right] - \widetilde{F}(\mathbf{w}^{(t,0)})
$$

$$
\leq \left(-\frac{m\tau\eta_l\eta_g}{m+\epsilon} + \frac{4\eta_l^3\eta_g\tau^3 L_s^2\beta'^2 m}{\epsilon} + \nu(\tau\eta_l\eta_g)^2\left(24L_s^2\eta_l^2\tau^2\beta'^2 + 3\right)\right)\left\|\nabla\widetilde{F}(\mathbf{w}^{(t,0)})\right\|^2 + (\tau\eta_l\eta_g)^2\nu\frac{\sigma_g^2}{m\tau}
$$

$$
+ (\tau\eta_l\eta_g)^2\nu\left(6L_s^2\eta_l^2\tau(\sigma_g^2 + 4\tau\kappa'^2) + \frac{12(M-m)L_c^2}{\tau^2 m(M-1)}\right) + \frac{4L_s^2\tau^3\eta_l^3 m\eta_g\kappa'^2}{\epsilon} + \frac{L_s^2\tau^2\eta_l^2 m\eta_g\sigma_g^2}{\epsilon} \tag{127}
$$

658 where $\nu = L_s + L_c/4$. With $\eta_l \leq 1/4\beta'\tau L_s$, $\epsilon = m$, and $\eta_g\eta_l \leq \frac{1}{9\tau\nu}$, we can further bound above
659 as

$$
\mathbb{E}\left[\widetilde{F}(\mathbf{w}^{(t+1,0)})\right] - \widetilde{F}(\mathbf{w}^{(t,0)}) \leq -\frac{\eta_l\eta_g\tau}{4}\left\|\nabla\widetilde{F}(\mathbf{w}^{(t,0)})\right\|^2 + (\tau\eta_l\eta_g)^2\nu\frac{\sigma_g^2}{m\tau}
$$

$$
+ (\tau\eta_l\eta_g)^2\nu\left(6L_s^2\eta_l^2\tau(\sigma_g^2 + 4\tau\kappa'^2) + \frac{12(M-m)L_c^2}{\tau^2 m(M-1)}\right) + 4L_s^2\tau^3\eta_l^3\eta_g\kappa'^2 + L_s^2\tau^2\eta_l^2\eta_g\sigma_g^2 \tag{128}
$$

660 Taking the average across all rounds on both sides of (128) and rearranging the terms we get

$$
\frac{1}{T}\sum_{t=0}^{T-1}\mathbb{E}\left[\|\nabla\widetilde{F}(\mathbf{w}^{(t,0)})\|^2\right] \leq \frac{4\left(\widetilde{F}(\mathbf{w}^{(0,0)}) - \widetilde{F}_{\inf}\right)}{T\eta_l\eta_g\tau} + 4\sigma_g^2\eta_l\left(\frac{\eta_g\nu}{m} + \frac{2L_s^2\eta_l\tau}{3} + L_s^2\tau\right)
$$

$$
+ \frac{80L_s^2\eta_l^2\tau^2\kappa'^2}{3} + \frac{48\eta_l\eta_g\nu(M-m)L_c^2}{\tau m(M-1)} \tag{129}
$$

661 With the small enough learning rate $\eta_l = 1/(\sqrt{T}\tau)$ and $\eta_g = \sqrt{\tau m}$ one can prove that

$$
\min_{t\in[T]}\mathbb{E}\left[\left\|\nabla\widetilde{F}(\mathbf{w}^{(t,0)})\right\|^2\right] \leq \frac{4\left(\widetilde{F}(\mathbf{w}^{(0,0)}) - \widetilde{F}_{\inf}\right) + 4\sigma_g^2\nu}{\sqrt{T\tau m}} + \frac{4\sigma_g^2 L_s^2}{\sqrt{T}} + \frac{8\sigma_g^2 L_s^2}{3\tau T}
$$

$$
+ \frac{80L_s^2\kappa'^2}{T} + \frac{48\nu(M-m)L_c^2\sqrt{\tau}}{\sqrt{T}m} \tag{130}
$$

$$
= \mathcal{O}\left(\frac{\sigma_g^2}{\sqrt{T\tau m}}\right) + \mathcal{O}\left(\frac{\sigma_g^2}{\tau T}\right) + \mathcal{O}\left(\frac{\kappa'^2}{T}\right) + \mathcal{O}\left(\frac{\sqrt{\tau}}{\sqrt{Tm}}\right) \tag{131}
$$

662 completing the proof for Theorem C.1 for partial client participation.

# E  Simulation Details for Fig. 4a

664 For the mean estimation simulation for Fig. 4(a), we set the true means for the two clients as
665 $\theta_1 = 0$, $\theta_2 = 2\gamma_G$ where $\gamma_G \in [0, \sqrt{20}]$. The simulation was perfomed using NumPy [41] and
666 SciPy [42]. The empirical means $\widehat{\theta}_1$ and $\widehat{\theta}_2$ are sampled from the distribution $\mathcal{N}(\theta_1, 1)$ and $\mathcal{N}(\theta_2, 1)$

respectively where the number of samples are assumed to be identical for simplicity. For local training we assume clients set their local models as their local empirical means which is analogous to clients performing a large number of local SGD steps to obtain the local minima of their empirical loss. For the global objective (standard FL, INCFL (ReLU), INCFL) a local minima is found using the `scipy.optimize` function in the SciPy package. For each $\gamma_G^2 \in [0, \sqrt{20}]$, the average IPR is calculated over 10000 runs for each global objective.

## F  Experiment Details and Additional Results

All experiments are conducted on clusters equipped with one NVIDIA TitanX GPU. The algorithms are implemented in PyTorch 1. 11. 0. All experiments are run with 3 different random seeds and the average performance with the standard deviation is shown. The code used for all experiments is included in the supplementary material.

### F.1  Experiment Details

**Obtaining $\widehat{\mathbf{w}}_i$, $i \in [M]$ for INCFL Results in Section 4.** In INCFL, we use $\widehat{\mathbf{w}}_i$, $i \in [M]$ to calculate the aggregating weights (see Algorithm 1). For all experiments with INCFL, we obtain $\widehat{\mathbf{w}}_i$, $i \in [M]$ at each client by each client taking 100 local SGD steps on its local dataset with its own separate local model before starting federated training. We use the same batch-size and learning rate used for the local training at clients done after we start the federated training (line 8-9 in Algorithm 1). The specific values are mentioned in the next paragraph.

**Local Training and Hyperparameters.** For all experiments, we do a grid search over the required hyperparameters to find the best performing ones. Specifically, we do a grid search over the learning rate: $\eta_l \eta_g \in \{0.1, 0.05, 0.01, 0.005, 0.001\}$, batchsize: $b \in \{32, 64, 128\}$, and local iterations: $\tau \in \{10, 30, 50\}$ to find the hyper-parameters with the highest test accuracy for each benchmark. For all benchmarks we use the best hyper-parameter for each benchmark after doing a grid search over feasible parameters referring to their source codes that are open-sourced. For a fair comparison across all benchmarks we do not use any learning rate decay or momentum.

**Logistic Regression on the Synthetic Dataset.** We conduct simulations on synthetic data which allows precise manipulation of heterogeneity. Using the methodology constructed in [2], we use the dataset with large data heterogeneity, `Synthetic(1,1)`. We have in total 100 devices where the local dataset sizes for each device follows the power law. The dimension used for logistic regression is $\mathbb{R}^{61 \times 10}$ where 10 is the output dimension.

**DNN Experiments.** For FMNIST, we train a deep multi-layer perceptron network with 2 hidden layers of units $[64, 30]$ with dropout after the first hidden layer where the input is the normalized flattened image and the output is consisted of 10 units each of one of the 0-9 labels. For CIFAR10, we train a deep convolutional neural network with 2 convolutional layers with max pooling and 4 hidden fully connected linear layers of units $[120, 100, 84, 50]$. The input is the normalized flattened convolution output and the output is consisted of 10 units each of one of the 0-9 labels. For Sent140, we train a deep multi-layer perceptron network with 3 hidden layers of units $[128, 86, 30]$ with pre-trained 200D average-pooled GloVe embedding [43]. The input is the embedded 200D vector and the output is a binary classifier determining whether the tweet sentiment is positive or negative with labels 0 and 1 respectively. All clients have at least 50 data samples.

### F.2  Additional Experimental Results

**Local Tuning for Personalization.** Personalized FL methods can be used to fine-tune the global model at each client before comparing it with that client's locally trained model. INCFL can be combined with these methods by simply allowing clients to perform some fine-tuning iterations before computing the aggregation weights in Step 7 of Algorithm 1. Both for clients that are active during training and unseen test clients, we show in Table 2 that INCFL increases the fraction of incentivized clients by at least $10\%$ as compared to all baselines. For FMNIST, CIFAR10, and Sent140, the improvement in IPR over other methods is up to $27\%$, $39\%$, and $28\%$ respectively for active clients and $17\%$, $35\%$, and $4\%$ respectively for the unseen incoming clients.

Table 2: Incentivized participation rate (IPR) of locally-tuned models with 5 local steps from the final global models trained with different algorithms for seen clients and unseen clients (the corresponding preferred-model test accuracy is in Appendix F.2).

| | Seen Clients | | | Unseen Clients | | |
|---|---|---|---|---|---|---|
| | FMNIST | CIFAR10 | Sent140 | FMNIST | CIFAR10 | Sent140 |
| FedAvg | 0.38 (±0.06) | 0.19 (±0.07) | 0.25 (±0.09) | 0.39 (±0.06) | 0.20 (±0.07) | 0.42 (±0.06) |
| FedProx | 0.40 (±0.07) | 0.17 (±0.07) | 0.26 (±0.09) | 0.41 (±0.07) | 0.19 (±0.07) | 0.43 (±0.12) |
| PerFedAvg | 0.45 (±0.05) | 0.26 (±0.02) | 0.24 (±0.10) | 0.46 (±0.06) | 0.28 (±0.04) | 0.47 (±0.06) |
| MW-Fed | 0.28 (±0.07) | 0.01 (±0.01) | 0.08 (±0.01) | 0.39 (±0.04) | 0.06 (±0.03) | 0.20 (±0.01) |
| INCFL | **0.55** (±0.01) | **0.40** (±0.00) | **0.36** (±0.05) | **0.56** (±0.01) | **0.41** (±0.01) | **0.55** (±0.01) |

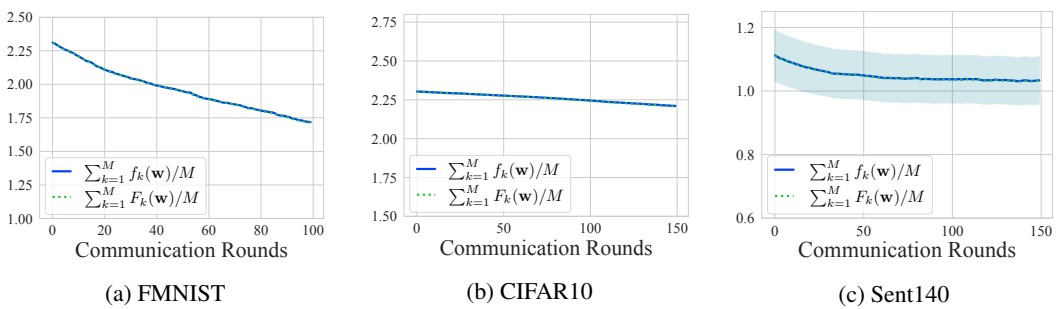

(a) FMNIST                    (b) CIFAR10                    (c) Sent140

Figure 7: Comparison of the average of the true local losses across all clients ($\sum_{k=1}^{M} f_k(\mathbf{w})/M$) and the empirical local losses across all clients ($\sum_{k=1}^{M} F_k(\mathbf{w})/M$) where the former is calculated on the test dataset and the latter is calculated on the training dataset for the global model $\mathbf{w}$. We show that the average of the true local losses is nearly identical to the average empirical local loss across all clients empirically validating our relaxation of replacing $f_k(\mathbf{w})$ with $F_k(\mathbf{w})$.

**Ablation Study on $f_k(\mathbf{w}) \approx F_k(\mathbf{w})$.** One of the two key relaxations we use for INCFL (see Section 2.1) is that we replace $f_k(\mathbf{w}) - f_k(\widehat{\mathbf{w}}_k)$ with $F_k(\mathbf{w}) - F_k(\widehat{\mathbf{w}}_k)$. In other words, we replace the true loss $f_k(\mathbf{w}) = \mathbb{E}_{\xi \sim \mathcal{D}_k}[\ell(\mathbf{w}, \xi)]$ with the empirical loss $F_k(\mathbf{w}) = \frac{1}{|\mathcal{B}_k|} \sum_{\xi \in \mathcal{B}_k} \ell(\mathbf{w}, \xi)$ for all clients $k \in [M]$. We have used the likely conjecture that the global model $\mathbf{w}$ is trained on the data of all clients, making it unlikely to overfit to the local data of any particular client, leading to $f_k(\mathbf{w}) \approx F_k(\mathbf{w})$. We show in Fig. 7 that this is indeed the case. For all DNN experiments, we show that the average true local loss across all clients, i.e., $\sum_{k=1}^{M} f_k(\mathbf{w})/M$ is nearly identical to the average empirical local loss across all clients, i.e., $\sum_{k=1}^{M} F_k(\mathbf{w})/M$ given the training of the global model $\mathbf{w}$ throughout the communication rounds. This empirically validates our relaxation of the true local losses to the empirical local losses.

**Preferred-model Test Accuracy for the Local-Tuning Results in Table 2.** In Table 2, we have shown how INCFL can largely increase the fraction of incentivized clients compared to the other baselines even when jointly used with local-tuning. In Table 3, we show the corresponding preferred-model test accuracies. We show that for the seen clients that were active during training, INCFL achieves at least the same or higher preferred-model test accuracy than the other methods for all the different datasets. Hence, the clients are able to also gain from INCFL by achieveing the highest accuracy in average with their preferred models (either global model or solo-trained local model). For the unseen clients with FMNIST, FedProx achieves a slightly higher preferred-model test accuracy (+0.05) than INCFL but with a much lower IPR of 0.46 (see Table 2) as INCFL's IPR is 0.56. For the other datasets with unseen clients, INCFL achieves at least the same or higher preferred-model test accuracy than the other methods. This demonstrates that INCFL consistently largely improves the IPR compared to the other methods while losing very little, if any, in terms of the preferred-model test accuracy.

**Comparison with Algorithms for Fairness** Fair FL methods [33, 34] aim in training a global model that yields small variance across the clients' test accuracies. These methods may incentivize the worst performing clients to participate, but potentially at the cost of disincentivizing the best performing clients. We show in Table 4 that the common fair FL methods are indeed not effective in

Table 3: Preferred-model test accuracy with the locally-tuned models with 5 local steps from the final global models trained with different algorithms for seen clients' and unseen clients' test data (the corresponding IPR is in Table 2).

| | Seen Clients | | | Unseen Clients | | |
|---|---|---|---|---|---|---|
| | FMNIST | CIFAR10 | Sent140 | FMNIST | CIFAR10 | Sent140 |
| FedAvg | 99.37 (±0.24) | 100.00 (±0.00) | 55.71 (±0.46) | 99.50 (±0.02) | 100.00 (±0.00) | 58.79 (±0.67) |
| FedProx | 99.35 (±0.23) | 100.00 (±0.00) | 55.75 (±0.80) | **99.55** (±0.09) | 100.00 (±0.00) | 58.82 (±0.72) |
| PerFedAvg | 99.20 (±0.25) | 100.00 (±0.00) | 55.74 (±0.80) | 98.98 (±0.55) | 100.00 (±0.00) | 58.82 (±0.72) |
| MW-Fed | 99.27 (±0.39) | 100.00 (±0.00) | 55.06 (±0.38) | 99.47 (±0.08) | 100.00 (±0.00) | 57.36 (±0.71) |
| INCFL | **99.40** (±0.30) | 100.00 (±0.00) | **55.82** (±0.82) | 99.50 (±0.02) | 100.00 (±0.00) | **58.88** (±0.77) |

Table 4: Incentivized participation rate (IPR) and preferred-model test accuracy for the seen clients' test data with the final global models trained via INCFL and q-FFL [33] which aims in improving fairness. The baseline q-FFL with large $q$, e.g. $q = 10$, emulates the behavior of another well-known algorithm for improving fairness named AFL [34].

| | Incentivized Participation Rate (IPR) | | | Preferred-Model Test Acc. | | |
|---|---|---|---|---|---|---|
| | FMNIST | CIFAR10 | Sent140 | FMNIST | CIFAR10 | Sent140 |
| q-FFL ($q = 1$) | 0.03 (±0.01) | 0.00 (±0.00) | 0.09 (±0.06) | 99.24 (±0.05) | 100.00 (±0.00) | 53.10 (±2.63) |
| q-FFL ($q = 10$) | 0.00 (±0.00) | 0.00 (±0.00) | 0.09 (±0.00) | 98.90 (±0.01) | 100.00 (±0.00) | 52.71 (±1.40) |
| INCFL | **0.55** (±0.00) | **0.40** (±0.00) | **0.41** (±0.07) | **99.29** (±0.03) | 100.00 (±0.00) | **53.93** (±1.87) |

improving the overall clients' incentivized participation rate. We see that the fair FL methods achieve an incentivized participation rate lower than 0.01 for all datasets while INCFL achieves at least 0.40 for all datasets. Moreover, the preferred-model test accuracy is also higher for INCFL compared to the fair FL methods. This underwelming performance of fair FL methods in incentivizing clients can be due to the fact that fair FL methods try to find the global model that performs well, in overall, over *all* clients which results in failing to incentivize *any* client.

