# OpenReview forum: "To Federate or Not To Federate: Incentivizing Client Participation in Federated Learning"
_NeurIPS.cc/2022/Workshop/Federated_Learning — FL-NeurIPS 2022 Oral_

### Official Review · Reviewer_GQaC · 2022-10-04
**Well written paper with an interesting idea, but few important limitations**

The paper introduces a new objective for federated learning, where the trained global model aims to perform better than the local alternative for as many clients as possible. The authors introduce an elegant solution to solve this new objective that is compatible with existing FL frameworks.

Few limitations that I would suggest for authors to discuss/address.
1. It seems that your premise is that each client only participates if its local model is worse than the global model. But to know this, all clients need to participate in training, which goes against incentives, i.e., each client has to participate anyway regardless of whether it is incentives or not. Why not then directly minimize global loss?
2. Follow up on my first comment. Based on your loss function, the global model is only incentivized to be somehow better than the global. Still, it can't guarantee that the global model will lead to very high accuracy, e.g., of a centralized model.

---

### Official Review · Reviewer_FHQ3 · 2022-10-09
**interesting and timely topic in federated learning**

This paper explores how to actively incentivize clients in FL. By a new definition of incentivized participation rate and corresponding formulation, they propose an algorithm called INCFL that dynamically adjusts the aggregation weights assigned to their updates. Concrete examples and convergence analysis are provided in the appendix.

pros:
1.	The notion of Incentivized Participation Rate is novel, and the approximation is reasonable.
2.	Iterative algorithm like FedAvg is proposed with theoretical analysis.

cons:
1.	The practical INCFL solver involves calculating the loss over its full training data for client, which is a heavy burden.
2.	In essence, the client participation is still assumed as independently random. I believe it would be more interesting if we considered cases when client participation depends on incentives.
3.	I am a little bit confused about the explanation of the aggregating weight q. For large or small incentive gap, q is nearly 0, but q is large for incentive gap near 0.

In general, this paper proposes a new notion of incentive and new objective function. It is interesting and timely. I believe this would be beneficial for the FL community.

---

### Decision · Program_Chairs · 2022-10-20

Accept (Oral)